# Enabling remote quantum emission in 2D semiconductors via porous metallic networks

Jose J. Fonseca[1]*, Andrew L. Yeats [1], Brandon Blue[2], Maxim K. Zalalutdinov[1], Todd Brintlinger[1], Blake S. Simpkins [1], Daniel C. Ratchford [1], James C. Culbertson[1], Joel Q. Grim [1], Samuel G. Carter [1], Masa Ishigami[2], Rhonda M. Stroud [1], Cory D. Cress [1] & Jeremy T. Robinson [1]*

Here we report how two-dimensional crystal (2DC) overlayers influence the recrystallization of relatively thick metal films and the subsequent synergetic benefits this provides for coupling surface plasmon-polaritons (SPPs) to photon emission in 2D semiconductors. We show that annealing 2DC/Au films on $SiO_2$ results in a reverse epitaxial process where initially nanocrystalline Au films gain texture, crystallographically orient with the 2D crystal overlayer, and form an oriented porous metallic network (OPEN) structure in which the 2DC can suspend above or coat the inside of the metal pores. Both laser excitation and exciton recombination in the 2DC semiconductor launch propagating SPPs in the OPEN film. Energy in-/out- coupling occurs at metal pore sites, alleviating the need for dielectric spacers between the metal and 2DC layer. At low temperatures, single-photon emitters (SPEs) are present across an OPEN-$WSe_2$ film, and we demonstrate remote SPP-mediated excitation of SPEs at a distance of 17 μm.

[1] U.S. Naval Research Laboratory, Washington, DC 20375, USA. [2] Department of Physics and Nanoscience Technology Center, University of Central Florida, Orlando, FL 32816, USA. *email: jose.fonsecavega.ctr@nrl.navy.mil; jeremy.robinson@nrl.navy.mil

ntegrated quantum optics platforms using solid-state quantum emitters have motivated the search for ideal on-chip single-photon sources that can be interfaced with one another to form networks for quantum information technologies. 2D semiconductors may provide such sources, with facile integration of on-chip elements due to their intrinsic immunity to total internal reflection and the possibility of van der Waals epitaxy on a wide range of substrates. The initial discovery of random single-photon emitters (SPEs) in WSe₂[1–5] quickly led to the deterministic placement of SPEs with narrow linewidths[6–9]. While the exact nature of the emitter sites remains unknown, the general consensus is that both defects[2–5,10] and strain[11,12] are sufficient to localize excitons with very narrow radiative linewidths. When paired with metals, excitons in a 2D semiconductor can be advantageously utilized via coupling into surface plasmon-polariton (SPPs) modes and vice versa[13,14]. This opens opportunities for energy routing between SPE sites in 2D semiconductors for applications such as on-chip quantum light sources[15–17]. In addition, metal nanostructures can lead to enhanced Purcell factors resulting in increased single-photon emission rates[9,18]. In each of these cases, the quality of the 2DC/metal interface will play a central role in determining the ultimate performance of the device. A generalizable technique to fashion high-quality epitaxial 2DC/metal junctions could enable a wide range of applications for 2DCs in quantum information science and photonics.

Surface and interface quality are often decisive factors in the performance of engineered solid-state devices. In the case of 2DC/metal junctions, assembly by transfer stacking (as opposed to direct metal deposition) reduces both chemical disorder and Fermi level pinning at the interface. Such devices can approach the Schottky–Mott limit for a van der Waals metal–semiconductor junction[19]. Alternatively, 2DC/metal interfaces with crystallographic registry can be formed through van der Waals epitaxy[20]. During physical vapor deposition, thin metal films tend to preferentially align on 2DCs[21,22], which results in periodic (Moiré) structural and electronic variations at the interface due to the lattice mismatch. Here, we demonstrate a material platform that brings together concepts from each of these areas: aligned 2DC/metal interfaces, SPP-exciton coupling, and 2D quantum emitters.

In this work, we exploit dewetting of 2DC/metal films to assemble structures that can be used for energy propagation and energy transfer between a 2D semiconductor and a metal film. The relative surface energy of a thin metal film on a substrate determines the driving force for wetting or dewetting[23]. Despite being atomically thin, 2D materials significantly modify metal film dewetting due to their relatively high Young's Modulus, which suppresses surface fluctuations[24]. We show here that under appropriate annealing conditions, the metal underlayer becomes highly textured and in crystallographic alignment with the 2DC overlayer, analogous to a reverse epitaxy process (or surface-templated epitaxy). As dewetting initiates, pores form in the metal film while the 2DC overlayer either remains suspended above or coats the inside of the pores. We term this 2DC/metal framework an Oriented Porous mEtallic Network (OPEN) film. In this OPEN film geometry, simultaneous energy coupling into SPPs at pore sites occurs via the direct laser field and via the decay of excitons in the 2D semiconductor. The existence of pores enables significantly improved SPP coupling from in-plane exciton dipole moments (and vice versa), which do not in-couple/out-couple well on planar metal regions[14]. We subsequently observe energy out-coupling in the form of remote photoluminescence (PL), the intensity of which is well described by a SPP decay length for each excitation wavelength. Furthermore, the porous metallic framework connects neighboring emitter sites located in the pores, creating an interconnected network of quantum emitters.

## Results

**2DC/metal architectures**. The basic structure for our 2DC/metal systems consist of a 2DC layer transferred onto a thin metal film without the use of a metal adhesion layer (Fig. 1a and Methods section). The simplicity of the process allows for the use of various 2DCs and metals in a wide variety of combinations. In this work, we focus on Au thin films on SiO₂ with a variety of 2DC overlayers, but the process appears to be general, working with other metal films (Supplementary Fig. 1). The Au film properties, substrate and annealing conditions were selected to promote metal dewetting at low temperatures even in the absence of the 2DC layer. We find the addition of hydrogen (H₂) gas to the annealing ambient reduces the onset temperature of dewetting by at least 100°C as compared to Ar alone (Fig. 1b, c), presumably through the reduction of suboxides and/or local carbonaceous pinning sites.

The dramatic effect a 2DC overlayer has on Au dewetting is highlighted in Fig. 1d. In this example, a narrow exfoliated strip of few-layer graphene was deposited on a 25 nm thick Au film and the sample was annealed at 350 °C for 30 min As thermodynamic modeling suggests[24], Au film dewetting is impeded beneath the graphene layer as compared with the Au-only region. Unlike the full prevention of Au dewetting reported by Cao et al.[24], we find reduced dewetting under graphene, resulting in metal pores (dark regions, Fig. 1d) that range in diameter from ~100–1000 nm depending on annealing conditions. An important difference between our work and that of Cao et al.[24] is the use of hydrogen gas during annealing. Atomic force microscopy (AFM) of this and other samples (e.g., Fig. 1e) confirm that the 2DC layers can remain suspended above the metal pores with close to 100% yield (forming tens-of-thousands of nanomechanical membranes in one step, Supplementary Fig. 2), despite the strain involved as the metal underlayer moves. Gold diffusion during dewetting leads to local variations in film height of up to about ±5 nm over large distances (dependent on the extent of pore coverage), which influences the resultant shape of and tension within the suspended 2DC membranes (Fig. 1e, f). As shown in Fig. 1f, the suspended membranes affix themselves to the top interior edge of the pore wall, which is commonly observed in strained 2DC membranes[25].

The 2DC overlayer not only regulates the metal dewetting process, it strongly affects the recrystallization of the Au film itself. As seen in the AFM image in Fig. 1e, the Au underlayer forms faceted pores and the Au surface has clearly resolvable step boundaries, suggesting a transformation from a nano- to microcrystalline film. Atomic scale imaging shows these 2DC/Au terraces are atomically smooth (RMS roughness ~10 pm), with terrace-to-terrace roughness on the order of 1 nm (Supplementary Figs. 3 and 4). Using electron backscattered diffraction (EBSD; see Methods) we can assess how the Au film recrystallizes with and without a 2DC overlayer. Figure 1g (and Supplementary Fig. 5) shows a {111} pole plot from two annealed samples: Au/SiO₂ and monolayer-MoS₂/Au/SiO₂. We specifically choose exfoliated material here to have confidence of a single-crystal 2DC domain across the entire analyzed area (60 × 40 μm²) and note that EBSD is insensitive to the 2DC monolayer. The pole plots show that the Au film becomes highly textured in the presence of MoS₂ after annealing (300 °C, 1 h), with two {111} in-plane rotations offset by 60° (Supplementary Fig. 6). While EBSD cannot inform on the relative orientation of the MoS₂ lattice to the Au lattice, we repeated the sample fabrication on a TEM grid to obtain the Fast Fourier transformation (FFT) diffractogram of a high-angle-annular-dark-field scanning transmission-electron microscope (HAADF-STEM) image (Fig. 1h). The boundary between suspended MoS₂ and MoS₂/Au shows the relative crystallographic alignment of the two crystals. For this sample,

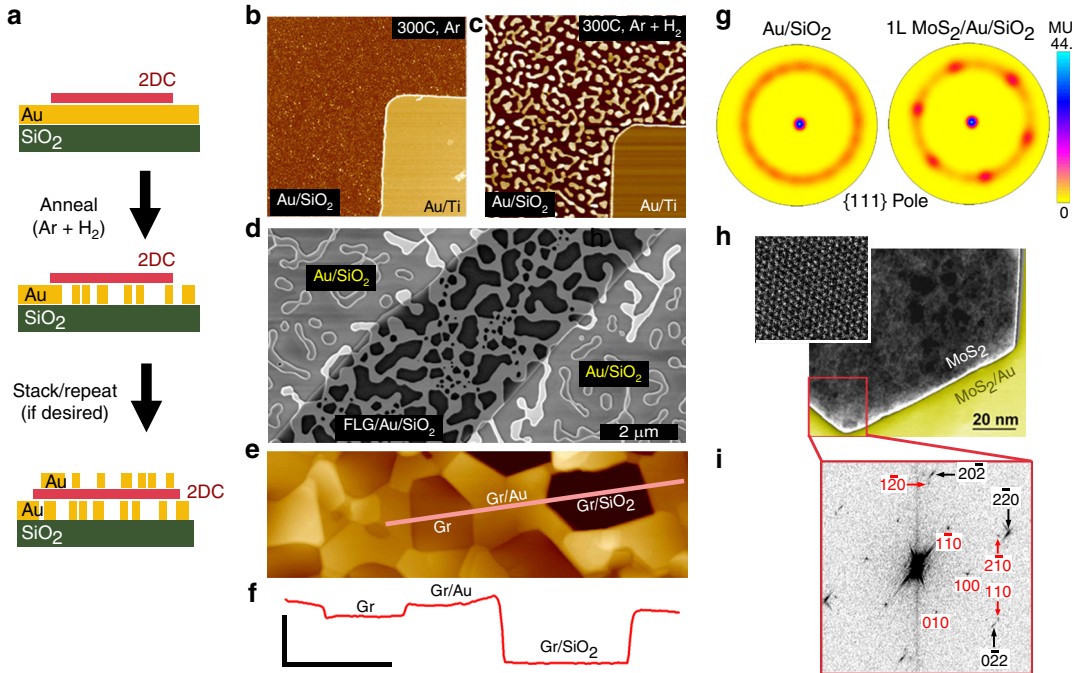

**Fig. 1 Fabrication & characterization of Au films with 2DC overlayers. a** Schematic of sample fabrication and processing. **b, c** AFM images showing Au+ patterned Au/Ti layers on SiO$_2$ after annealing in flowing **b** Ar and **c** Ar + H$_2$ at 300 °C (image: 14 × 14 μm$^2$). **d** SEM image of an exfoliated few-layer graphene (FLG) flake on Au/SiO$_2$ after annealing. **e** AFM image of a CVD graphene (Gr) monolayer on de-wetted Au pores. **f** Cross-sectional height profile from the line drawn in **e** showing an example of a suspended graphene membrane and a graphene membrane coating the inside of a pore (scale bars: horizontal = 500 nm, vertical = 30 nm). **g** EBSD {111} pole figures from annealed Au/ SiO$_2$ and annealed monolayer-MoS$_2$/Au/SiO$_2$ (scale = multiple of uniform density (MUD)). **h** STEM-HAADF imaging and (**i**) corresponding FFT diffractogram showing the relative lattice orientation of a MoS$_2$ membrane (red indices) anchored to a de-wetted Au support (black indices).

the relative lattice registry of all MoS$_2$-Au boundaries at various pore edges was within ±4°. To our knowledge, similarly faceted metallic pores supporting crystallographically aligned, suspended 2DC membranes have not been achieved using conventional lithography/etching and 2DC film transfer techniques.

By closely monitoring Au dewetting beneath a 2DC layer, we gain insight into the dewetting dynamics. The evolution of metallic pore formation and growth under a WS$_2$ monolayer during cumulative annealing is illustrated in Fig. 2a, b (optical image of the same region shown in Fig. 3d). The sequential annealing and imaging of the same location enables a quantitative analysis of changes in pore size, shape and frequency (or count) (Fig. 2c and Supplementary Figs. 7 and 8). We observe here that most of the pore formation and growth occurs within the first 60 min of treatment. After an initial 15 min anneal (300 °C) there is already a large density of small pores (<Area> = 0.028 μm$^2$, density = 3.13 μm$^{-2}$), with a surface coverage of 8.9%. With further annealing, pores tend to grow, merge and exhibit well-defined facets, while the top Au surface is smooth with distinct step edges. At the longest anneal times (360 min) for this sample, the pore coverage increases to 29.6%, the average pore size increases by 5.5× (<Area> = 0.154 μm$^2$), and the pore density decreases by ~30% (density = 2.17 μm$^{-2}$). Figure 2b highlights an example of these changes, where we outline individual pores circled in Fig. 2a as they change with cumulative annealing.

The growth of larger metal pores, seemingly at the expense of smaller ones (e.g., Fig. 2b), resembles a behavior analogous to Ostwald ripening[26] of solid particles. In this case, however, the particles under consideration are pores (i.e., absence of material). One of the more successful coarsening (or ripening) models—the so-called LSW theory named after Lifshitz and Slyozov[27], as well as the work of Wagner[28]—uses a log-normal distribution to follow how particles lose or gain mass over time. When we apply

a LSW model (Fig. 2c, solid lines) to the pore distributions, we find relatively good overlap (R$^2$ values ranging from 0.887 to 0.977) to the data, which suggests pore growth here follows conventional particle-ripening kinetics. As mentioned earlier, the competing forces at play are the differences in surface energy between the metal-substrate interface and the metal-2DC interface, as well as the high Young's modulus of the 2DC, which reduces surface fluctuations that initiate dewetting[24]. Without the 2DC overlayer, the Au film completely dewets to form localized Au islands with randomly oriented in-plane crystal directions.

It is also important to note the impact of defects in the 2DC layer on Au dewetting. Figure 2d shows an optical image of CVD graphene/Au film that has been selectively treated with oxygen plasma (15 sec) to intentionally introduce defects in the top graphene layer. When subsequently annealed, Au beneath pristine graphene dewets as expected (bluish region, Fig. 2d), whereas Au dewetting beneath defective graphene is more limited. We surmise that dangling bonds and functional groups in the graphene layer can effectively serve as pinning sites for Au diffusion, as well as change the local surface energy, and thereby impede the dewetting process. Using this result, we have selectively patterned the wetting/dewetting behavior of an underlying Au film with few-micron resolution (inset Fig. 2d and Supplementary Fig. 9).

**Optical and electronic properties of 2DC/metal architectures.** Beyond morphological considerations, the OPEN film geometry leads to a rich mixture of electronic and optical properties across the 2DC/metal layers. Specifically, a discrete distribution of nano- and micron-scale 2DC/metal interfaces form after dewetting. They consist of: (i) planar, crystallographically aligned 2DC/metal boundaries, (ii) sharp metallic pore edges and corners with a

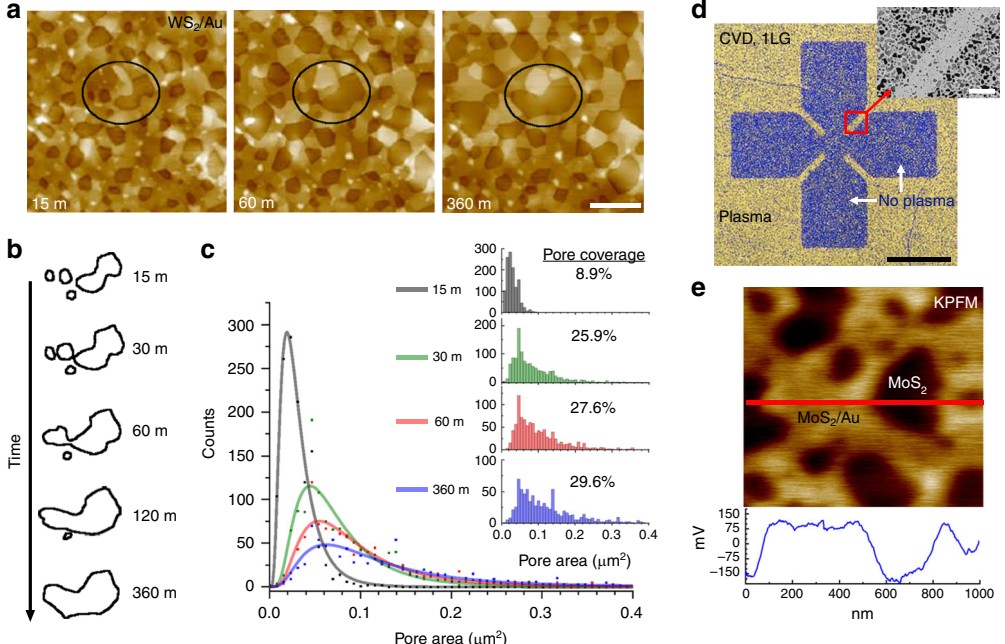

**Fig. 2 Evolution of 2DC/Au/SiO₂ samples during annealing. a** Series of AFM images from the same monolayer-WS₂/Au region after different cumulative annealing times (Ar + H₂, 300 °C; scale bar = 1 μm) (note: optical images of this region shown in Fig. 3d). **b** Outline of the pores inside the black ovals in **a** with increasing annealing time. **c** Plot showing the histogram peak value of the pore area vs. count for different cumulative annealing times (processed using ImageJ[40]). The solid lines are fits using the LSW particle-ripening model[26]. (inset) Full histograms of the pore area vs. count for different cumulative annealing times, along with the measured areal pore coverage for each annealing time. **d** Optical image showing a CVD graphene/Au/SiO₂ sample that has been selectively treated with a 15 sec O₂-plasma (scale bar = 40 μm). After annealing, the Au layer beneath pristine graphene dewets, while the Au beneath O₂-plasma exposed graphene has limited dewetting (inset scale bar = 3 μm) **e** KPFM image of a MoS₂/Au sample after annealing. The suspended MoS₂ membranes can have up to a 300 mV potential difference compared to the Au-supported MoS₂.

conformal 2DC coating, and (iii) strained and suspended 2DC layers. It is known that hybridized electronic states form at metal/2D semiconductor junctions, where the bandstructure of the semiconductor typically renormalizes[29]. We confirm such changes exist here using kelvin probe force microscopy (KPFM) to measure the surface potential of an OPEN-MoS₂ film (Fig. 2e and Supplementary Fig. 10). The surface potential of the free-standing MoS₂ areas differs by up to 300 mV from the potential of the MoS₂ in contact with Au, consistent with other reports[30]. This variation primarily corresponds to band bending of MoS₂, due to the difference between the Au and MoS₂ work functions[30,31]. As a result, regions of suspended 2DC membranes in the OPEN film can serve as localized potential wells for charge carriers (e.g., electrons or holes depending on the metal/2DC junction) and exhibit a notable impact on emission properties, as described below.

Variations in the optical properties (both local and remote) across the OPEN film are equally striking and are the subject of the remaining analysis and discussions. Since light emission in 2D semiconductors is quenched on metal surfaces, we anticipate large intensity differences between the Au-supported and free-standing 2DC regions (Fig. 3a). Photoluminescence (PL) on an OPEN-WS₂ film and a supported WS₂/Au-Ti film shows that the free-standing OPEN-WS₂ membranes can have 1000× brighter PL than the Au-supported regions (Fig. 3c and Supplementary Fig. 11). When compared to a sample of WS₂ on PMMA, the peak PL intensity of the OPEN-WS₂ film is about three times larger (Supplementary Fig. 11). To visualize this on a larger scale, Fig. 3d shows room temperature optical and fluorescence images of a WS₂/Au sample before and after annealing. Local regions of Au/Ti were patterned here as anchoring sites (dewetting does not occur with a Ti sticking layer) to illustrate another avenue for directed pore formation. Immediately upon formation of an

OPEN-WS₂ film, we observe fluorescence that intensifies with increased annealing time as more free-standing WS₂ is generated (Fig. 3d lower right).

Remote excitation of PL, facilitated by SPP traveling waves in the metal framework, can also occur over tens of microns in these samples. Excitation transfer over these distances is shown in widefield fluorescence microscopy (Fig. 3e), where bright emission appears at pore sites up to about 20 μm from a fixed laser illumination spot (3.5 μm FWHM). Our findings support the presence of SPPs being responsible for this long-range excitation (schematically shown in Fig. 3b), which is similar to that observed in TMD/dielectric/metal geometries[13]. To characterize the SPPs that propagate in our materials, we image them directly with near-field scanning optical microscopy (NSOM) at room temperature (Fig. 3f). SPPs generated from free space excitation through a perforated metal film should produce interference patterns whose periodicity match the SPP's in-plane wavelength[32,33]. Considering the dielectric function of Au, $\varepsilon_m$[34], and incident excitation $\lambda_i = 532$ nm, the SPP's wavelength would be $\lambda_{SPP} \approx \lambda_i \sqrt{(\text{Re}(\varepsilon_m) + 1)/\text{Re}(\varepsilon_m)} = 471$ nm. This matches well with oscillatory fringes we observe in a section profile taken through Fig. 3f and plotted in Fig. 3g. As expected for fringes associated with SPPs emanating from a nanopore[32], fringes are more pronounced along the direction of incident polarization (horizontal polarization identified by the white arrow in Fig. 3f; vertical polarization response is included in the Supplementary Fig. 12). The propagation length (γ) of these 532 nm excited SPPs is dominated by resistive losses in the metal, and is estimated at γ ≈ 1 μm[32,35]. This direct NSOM imagining clearly identifies the presence of SPPs launched from the myriad holes present in the metal framework and provides the basis for remote excitation of bright emission centers.

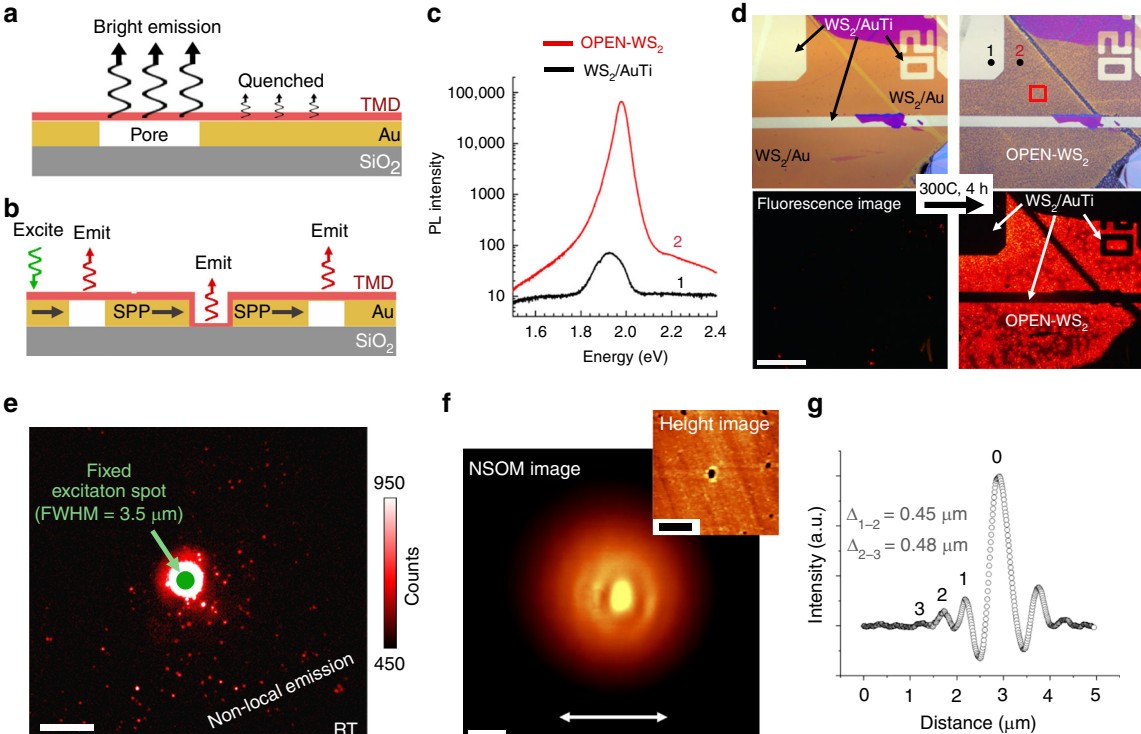

**Fig. 3 Optical characterization of OPEN-WS₂ samples.** Cartoon illustrating **a** strong emission from the suspended TMD and quenched emission where it is supported by Au and **b** remote excitation and emission from the an OPEN-WS₂ film. **c** PL spectra (log₁₀ intensity) taken from spot 1 and spot 2 in the optical image in **d**. **d** Optical and fluorescence images (integrated CCD image) of a monolayer-WS₂/Au/SiO₂ sample before and after annealing (red box = AFM region from Fig. 2a; scale bar = 50 μm). Regions of Au-Ti were selectively patterned to inhibit dewetting at specific locations. **e** Fluorescence image of an OPEN-WS₂ film on Au with a fixed excitation laser spot (beam FWHM ≈ 3.5 μm, power = 4.5 mW, room temperature (RT)) demonstrating remote excitation of suspended WS₂ pixels (truncated intensity scale; scale bar = 10 μm). **f** Topography and near-field scanning optical microscopy (NSOM) interference pattern originating from SPPs (scale bar = 1 μm; inset scale bar = 2 μm; arrow shows polarization direction). **g** Horizontal cross-section of near-field intensity taken from **f**.

**Local and remote quantum emission**. Photon-mediated interactions between discrete on-chip quantum light sources is the foundation of integrated quantum optics. We demonstrate key requirements toward this goal, including the remote excitation of single-photon emitters (SPEs). Using a sandwiched OPEN-WSe₂ film (Fig. 1a bottom and Supplementary Figs. 13 and 22), we measure a distribution of narrow emission centers and extract relevant statistics (Supplementary Fig. 14). Figure 4a shows an example of a low-temperature scanning confocal PL map where the excitation and collection spots are coincident (Supplementary Figs. 23 and 24). A histogram of emitter peak energies from this map has a peak centered at 1.625 eV (inset Fig. 4a). From this same region, we can collect remote PL maps to quantify aspects of remote excitation. In Fig. 4b we fix the collection spot at one emitter (central black dot) and scan the excitation spot to learn where energy is in-coupled, transmitted through the porous metal framework, and out-coupled via the emitter under observation. Given the non-periodic structure of the OPEN film, we could expect a range in transmission efficiency across the sample. Away from the center, the bright spots in Fig. 4b. occur at positions of the excitation beam in which the combination of energy in-coupling, SPP transmission, and out-coupling at the central emitter are most efficient.

Using this experimental geometry, we perform time-correlated single-photon counting measurements to compare emitter response when excited directly by the laser field, versus excitation by a propagating SPP. Figure 4c shows the measured 2nd-order photon correlation function $g^{(2)}(\Delta t)$ of a SPE that is directly excited with 25 nW of laser power. The antibunching

dip below 0.5 indicates that it is a source of single photons, with a measured purity of $g^{(2)}_{x=0 \ \mu m}(\Delta t = 0) = 0.14$. The emitter can also be excited via SPPs propagating from a remote excitation spot. As shown in Fig. 4d, we find that the emitter remains a single-photon source when excited remotely by 10 μW of laser power at an in-coupling spot 17 μm away, in which case we measure a photon purity of $g^{(2)}_{x=17 \ \mu m}(\Delta t = 0) = 0.27$. Differences in the emitter lifetime, as estimated from the width of the $g^{(2)}(\Delta t)$ antibunching dip, are likely due to differences in the effective excitation power reaching the emitter site[36]. The individual emission spectra (Fig. 4e) for local (laser field) or remote (SPP) excitation reveals no significant differences in overall spectral shape. This result implies that a specific emitter (within a pore) that can be excited by the direct laser field, can also be excited by a propagating SPP interacting with the pore.

To help understand this phenomenon, we performed COMSOL simulations which show the coupling between an in-plane dipole (e.g., the bright exciton in WSe₂) and a SPP mode is increased by ~3–4 orders of magnitude due to the presence of a pore (pore-related hot spots and dipole-SPP coupling calculations shown in Supplementary Figs. 15–17). In particular, these simulations identify that emitter sites located at the pore edges would interact most strongly with a SPP and it is this dramatically enhanced interaction that is likely responsible for our observed SPP-pumped PL. Together, these simulation results help in visualizing how a propagating SPP can excite an exciton dipole emitter embedded within a metallic pore.

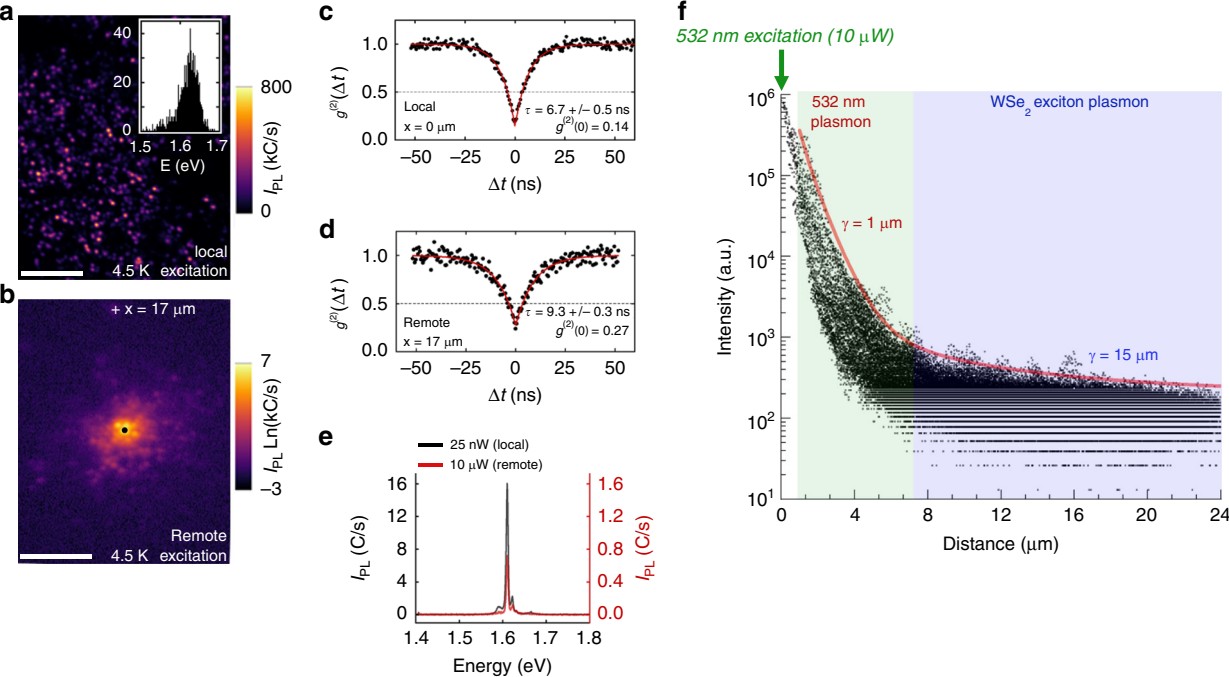

**Fig. 4 Optical characterization of a sandwiched OPEN-WSe₂ film.** Integrated intensity from the same area of a Au/monolayer-WSe₂/Au/SiO₂ sample using **a** scanning confocal PL map (linear color scale) and **b** remote PL map (logarithmic color scale) with the collection spot fixed at the black dot point and the excitation beam scanned to form the image (scale bar = 10μm). (inset **a**) histogram of dominant emitter peak energy versus number of emitters in **a**. **c**, **d** second-order photon correlation function $g^{(2)}(t)$ measurement of the central emitter in **b** for excitation at the black circle point and remotely at white cross point, respectively. **e** PL spectra taken at the central emitter while exciting locally (excitation and collection at the black dot) and remotely (excitation at the white cross, collection at the black dot). The scale of the remote spectrum (red curve) is on the right-hand y-axis. **f** Plot of PL intensity versus distance for every pixel in **b**. A fit to the most intense (peak intensity) PL data using Equation 1 (red line) has a $R^2 = 0.92$, with $\gamma_{532nm} = 1\,\mu m$ and $\gamma_{WSe2} = 15\,\mu m$ (Supplementary Fig. 19).

By plotting the intensity of every pixel versus distance from the central emitter in Fig. 4b (Fig. 4f and Supplementary Fig. 18), we can estimate the decay length of remote PL across the OPEN film. Notably, two distinct regions (slopes) are evident in the PL intensity plot. Since there are two pathways in which SPPs are excited in our samples: (i) by the polarized laser field via pores in the metal (532 nm excitation) and (ii) by exciton recombination in the 2D semiconductor, the total PL decay as a function of distance should be described by

$$I = I_b + I_{532\,nm}\frac{1}{x}e^{\frac{-x}{\gamma_{532\,nm}}} + I_{727\,nm}\frac{1}{x}e^{\frac{-x}{\gamma_{727\,nm}}} \quad (1)$$

where $I_\lambda$ is the initial wavelength-specific intensity at the launching site, $\gamma_\lambda$ is the wavelength-specific decay length, and $I_b$ is the background signal (see Methods). We note that the $\gamma_\lambda$ value here includes in it all loss mechanisms in the OPEN film, which include Ohmic loss, roughness, pore transmission, and inhomogeneties. The additional $1/x$ term arises from SPPs radiating in all directions in the 2D plane[13]. Fitting equation (1) to the most intense PL data from the SPE in Fig. 4f results in a good agreement ($R^2 = 0.92$) over three orders of magnitude (Supplementary Fig. 19). At less than about 6 μm from the emitter, the PL intensity decays rapidly and is fit well with $\gamma = 1\,\mu m$. Beyond 6 μm from the emitter, the decay rate slows and is fit well with $\gamma = 15\,\mu m$. As discussed earlier, the estimated decay length for plasmons in porous Au excited by 532 nm light is $\gamma \approx 1\,\mu m$, which provides confidence in assigning the initial decay constant as $\gamma_{532nm}$. The longer decay constant of $\gamma = 15\,\mu m$ can then be assigned as that due to excitons in the 2D semiconductor. In WSe₂, both neutral (X₀) and charged excitons (X_T) have a temperature-dependent energy that varies from ~750 nm to 710 nm (1.65–1.75 eV) from room temperature down to 4 K[37].

The Au-induced doping in WSe₂ (analogous to Fig. 2e) will result in higher emission intensity from X_T (versus X₀) and therefore, we can reasonably assign the most intense exciton dipole energy at ~λ = 727 nm at 4.5 K[14].

The good fit of Eq. (1) to the most intense PL data gives insight into the highest efficiency exciton−SPP−exciton pathways within the OPEN film structure. For example, the fit of the data for distances <6 μm yields a decay length $\gamma_{532nm} = 1\,\mu m$, which is consistent with decay dominated by Ohmic losses (i.e., pore scattering is not a significant contributor to SPP decay). On the other hand, if one maintains the assumption of only Ohmic losses and examines decay at λ = 727 nm, analytical treatments predict $\gamma_{727nm} = 32\,\mu m$ (Supplementary Fig. 20). Our fitted value ($\gamma_{727nm} = 15\,\mu m$) is 53% smaller, implying that there are additional SPP scattering mechanisms at longer wavelengths. We have performed COMSOL simulations (Supplementary Fig. 21) to gain a qualitative assessment of how the SPP propagation varies in the presence of pores and find two important results: (i) a decrease in $\gamma_{727nm}$ of ~55% due to pores of similar diameter and density as measured in Fig. 4 and (ii) approximately no difference in $\gamma_{532nm}$ when pores are included. Both results are in good qualitative agreement with the experimental observations here and support our contention that propagating plasmons, at λ = 532 and 727 nm, are present and account for remote excitation of PL.

While the presence of pores might initially appear as a significant scattering source, we note that SPPs can efficiently transmit their energy across gaps, especially at length scales less than a few hundred nanometers[38]. The average pore diameter in Fig. 4 is 230 nm (Supplementary Fig. 13), which would approximately correspond to a SPP transmission rate (T) of 80% if modeled as perfect 230 nm gap[38]. Since we have a

continuous metallic film, the transmission rates from Flynn et al.[38] can serve as a lower bound for approximating loss due to pores. Assuming the additional 727 nm SPP loss is from pore scattering alone, we have estimated a SPP pore transmission efficiency for this sample at T = 94% (Supplementary Fig. 13). This is ~18% larger than that found in Flynn et al. for perfect 230 nm gaps[38], suggesting an ideal pore size that both couples exciton dipoles and that supports SPPs with good transmission rates is less than or equal to a few hundred nanometers. Overall, this assessment provides confidence in the possibility of designing coupled metal/2DC platforms using the OPEN film concept, where pores with embedded 2D layers enable energy in-coupling/out-coupling while not being detrimental to SPP transmission efficiencies.

## Discussion

We have reported on the physical and electronic interplay between two-dimensional crystal (2DC) layers and self-assembled, highly textured porous metallic frameworks. Annealing a 2DC/Au/SiO$_2$ stack results in a reverse epitaxy process of the encapsulated Au layer, where the Au layer crystallographically orders, in registry, with the 2DC overlayer and transforms into an oriented porous metallic network (OPEN) film. The 2DC overlayer remains suspended above or coats the inside of the metal pores, which serve as local windows to access intrinsic properties of the 2DC layer. When using a semiconducting 2DC layer, bright emission occurs at the 2DC/metal-pore sites and is quenched at the 2DC/metal regions. The porous metal framework supports propagating surface plasmon-polaritons (SPPs) launched by either a direct laser field or from exciton decay in the TMD semiconductor. These SPPs travel tens of microns in the metallic framework to re-excite excitons in a TMD overlayer. Using this process, we measure both local and remote single-photon emission from the same emitter site, which shows the quantum nature of the emission is preserved whether the excitation originates from a laser field or a propagating SPP.

## Methods

**Sample fabrication**. To generate exfoliated 2DC/Au samples, Au films were first evaporated (thickness = 25 nm, pressure ≤ 8E-7 torr; rate = 1Ang s$^{-1}$, room temperature) onto PVA-coated Si substrates (PVA is used here as a sacrificial layer). Bulk crystals (graphene, h-BN, MoS$_2$, WS$_2$, and WSe$_2$; sourced from HQ graphene and 2D Semiconductors) were then exfoliated onto the Au/PVA/Si substrates to produce ultra large-area monolayer regions (>100 × 100 μm$^2$), similar to Desai et al.[39] A macroscopic scratch was drawn through the 2DC/Au/PVA layer around regions of interest and then local water drops were added to dissolve the PVA layer. The released 2DC/Au films were re-deposited on SiO$_2$/Si substrates. Large-area graphene films were grown by low-pressure CVD using Cu foils at 1030 °C with flowing H$_2$/CH$_4$ gas (total pressure between 20 and 100 mtorr) and subsequently transferred onto Au/SiO$_2$ substrates using wet chemical etching and a PMMA support coating. 2DC/Au film samples were subsequently annealed in a 1-inch clamshell tube furnace with flowing Ar (600 sccm) and H$_2$ (400 sccm) gas between 15–360 min, then quenched by opening the furnace.

**Near-field scanning optical microscopy (NSOM)**. NSOM measurements were carried out on a Witec alpha300 in collection mode. Light (532 nm laser, power = 0.23 mW) is incident through the bottom of the sample. The illumination optic is fixed to the scanning stage so that the illuminated area does not change during scanning. Light is collected through an apertured AFM tip (~90 nm aperture size, contact mode) and measured with an avalanche photodetector (APD).

**Variable-temperature confocal photoluminescence**. Variable-temperature PL measurements (4.5 K—RT) were acquired in a closed cycle optical cryostat with an internally mounted 100×, 0.85 NA microscope objective (Supplementary Fig. 24). A 532 nm diode-pumped solid-state laser was used for excitation. Emitted light from the sample was passed through a 600 nm long-pass filter and then coupled into an optical fiber that is routed either to a spectrometer or to a beamsplitter and two avalanche photodiodes (APDs) for photon correlation measurements. Scanning confocal PL images and split excitation-collection PL images were collected using two dual-axis scanning mirrors and two relay lens pairs (Supplementary Fig. 24), allowing separate control of both the laser excitation and confocal collection points

on the sample. When measuring with the APDs, we estimate the background count rate, I$_b$, at ~200 C/s. This rate is a combination of both the dark counts of the detector and the sample-independent background from our apparatus.

**Electron backscatter diffraction (EBSD)**. EBSD was carried out in a Helios Focused Ion Beam system. For the averaged diffraction plots in Fig. 1g, data were collected from 60 × 40 μm$^2$ areas on each region (same sample). Diffraction patterns were acquired approximately every 1.3 μm, totaling ~1200 data points.

**Scanning transmission-electron microscopy**. High-angle-annular-dark-field (HAADF) scanning transmission-electron-microscope (STEM) images were acquired with an aberration-corrected STEM (Nion UltraSTEM200X) operated at 60 kV with a probe current of about 50 pA. The STEM samples were prepared by transferring 2DC/Au films to commercial heating substrates coated with thin carbon films that contain arrays of 2 μm holes (Protochips E-FHBC). The as-transferred 2DC/Au films were then directly annealed on these substrates to form the OPEN film before insertion into microscope for imaging.

## Data availability

The authors declare that the data supporting this study are available within the article and its Supplementary Information files. Further information is also available from the corresponding authors upon reasonable request.

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

## Acknowledgements
This work was funded by the Office of Naval Research through Base Programs at the Naval Research Laboratory. MOCVD materials used in this study were provided by The Pennsylvania State University 2D Crystal Consortium—Materials Innovation Platform (2DCC-MIP) under NSF cooperative agreement DMR-1539916. This research was performed while J.J.F. held an NRC Research Associateship award at NRL.

## Author contributions
J.J.F. and J.T.R. conceived the idea for the project and fabricated samples. A.L.Y. performed and analyzed low-temperature spectroscopy measurements with help from S.G.C. and J.Q.G. Room temperature spectroscopy and AFM was performed by J.J.F., J.T.R, and J.C.C. AFM data were analyzed with ImageJ by B.B. with supervision from J.T.R. and M.I. STM measurements were performed and analyzed by B.B. with supervision from J.T.R. and M.I. Fluorescence microscopy was performed by J.J.F. STEM measurements were performed by T.B. and analyzed by T.B. and R.M.S. EBSD measurements were performed by M.Z. NSOM measurements and analysis were performed by B.S.S. and D.C.R. COMSOL simulations were performed by C.D.C. All authors discussed the results and commented on the manuscript.

## Competing interests
The authors declare no competing interests.
