## [Peer Review File · Nature Communications]

Reviewers' comments:

Reviewer #1 (Remarks to the Author):

The authors report on a hybrid system consisting of a porous metal film and a two-dimensional crystal. Authors demonstrate a technique to fabricate such system by annealing a thin and uniform metal film (Au, Ag, Pd and Ni) capped by a 2D crystal (WSe₂, WS₂, Gr, hBN, MoTe₂). It was demonstrated that 2D crystal alters the 'growth' dynamics (de-wetting) and crystallographic structure of the metal film itself. Authors have also shown that the formation of pores in a metal film capped by 2D crystal follows closely a LSW model.

The main result of the work is a demonstration of a dispersed distribution of single photon emitters achieved with a single laser spot excitation. Authors claim that the structure of the porous metal film enables propagation of surface plasmon-polaritons (SPPs) which then drives excitation of neighbouring single photon emitters (SPEs).

Please note that I avoid using the word 'non-local quantum emission' as in quantum optics field this often refers to non-locality property of entangled photons. I would suggest updating the title to avoid this ambiguity.

I found the evidence supporting some claims ambiguous and highly questionable. This mainly relates to the connection between SPP and SPE. There is also a big gap in interpretation of spectroscopic data (quality of SPEs, what about quantum well emission from WSe₂, what is exactly plotted in the PL maps).

Based on the above and on my more detailed comments below I recommend to reject the manuscript from publishing in the Nature Communications. In my opinion, the manuscript requires a lot of further work to become suitable. If of good quality, the further work might justify the resubmission.

Please see my comments below:

1. The main point of concern I have is that the authors do not provide strong enough evidence to support the claim that propagating SPP are responsible for exciting SPEs at different locations. The link between SPPs and SPEs is still unclear to me. Indeed, from NSOM measurements it is clearly shown that a single pore launches a SPP that propagates up to 2 μm . Although it is very important measurement, this does not capture the nature of SPP propagation in the porous film structure. How does SPP travel through porous metal film? How such short-travelling SPP can excite SPEs that are 17 μm away from the laser spot? Would cross-correlation ($g_2(\tau)$) measurement between two different 'non local' SPE provide an insight to involvement of SPP? A SPE that is further from the origin of SPP should in principle emit later than a SPE that is closer?

As demonstrated in Zhou Y. et. al. Nature Nanotechnology 12, 856 (2017), SPP couple to dark excitons in WSe₂. Do authors observe such enhancement of PL emission from dark excitons in monolayer WSe₂ enabled by OPEN?

Authors use work mentioned in ref. 13 and 36 to support their interpretation of the data. In ref. 13, it has been demonstrated that SPP can propagate up to 34 μm . This was achieved thanks to a smooth interface of 2D metal-oxide-semiconductor (Ag and WS₂). On the other hand, in this work porous morphology annealed metal films provides many out-coupling sites for SPP due to numerous terminations of metal layer which effectively create sharp dielectric changes between air and metal. From ref. 36: "The attenuation length is affected by the scattering of SPs on the surface roughness, scratches, and inhomogeneities of the metal film or the adjacent medium [therein 15,16]". Intuitively, this should decrease SPP propagation. Having this in mind I believe that it is unlikely that SPPs propagating along a porous metal are responsible for exciting SPEs at

distances on the order of 10s of μm .

There is an ambiguity in estimating SPP propagation length. On one hand, authors use the work from ref. 36 to give the length of $15\ \mu\text{m}$ for $750\ \text{nm}$ plasmons, where the influence of metal roughness is studied. On the other, authors use a formula for Ohmic losses (ref), (mentioned in the section 8 in the supplementary) which assumes flat and smooth interface to obtain $1\ \mu\text{m}$ propagation length for $532\ \text{nm}$ plasmons. In Fig.4f Plotted lines for γ do not reflect well the data. Are three lines fitted values or fixed values? What are the fitted values for decays? Could authors improve their strategy for estimating, measuring and fitting SPP propagation length?

As the skin depth of SPP is larger than the thickness of metal film used, SPP propagation along OPEN would be modulated by the roughness proportional to the given depth of sharply terminated pores ($25\ \text{nm}$) and their density. How the roughness of OPEN film is taken into account to estimate SPP propagation length?

The SPPs propagate in directions determined by the polarization of the incident beam. If the claims in the manuscript are to be accurate, I would expect that different polarisations should yield to different intensity distribution of PL maps that are fostered by OPEN. So, why the PL map in fig. 4b does not reflect this property of SPP and instead the PL map seems to have a circular symmetry which one would expect to observe when exciting with a circularly polarised light?

This makes me think that there might be another mechanism that enables the dispersed excitation of SPEs. An alternative effect could be that the metal surface with its numerous holes, sharp edges and variations in height scatters efficiently excitation light in all directions which could optically and directly excite neighbouring SPEs. This seems plausible to me since the sufficient excitation power for observing SPEs is in the nW range, as shown in fig. 4e. Moreover, to acquire the PL map shown in fig.3e, $4.5\ \text{mW}$ of pump power was used which is remarkably high amount of power, suggesting that the scattering might play a role. Could authors attempt to disprove or confirm this hypothesis?

If it is true that SPPs excite SPEs at neighbouring pores sites, I would expect that the intensity distribution of PL maps from fig.3e and fig.4b should look similar. However, these PL maps appear to look very different. Why is that? How would a histogram of peak intensity vs distance (as shown in Fig. 4f) would look like for PL map from Fig. 3e? Why not integrated intensity?

In such systems like OPEN-WSe₂ one could expect Purcell enhancement to be relevant and pronounced in SPE. Do authors observe such effects? Is this surprising that the 'non-local' emitter from fig. 4b,d,e shows even longer decay time than 'local' emitter? Can authors measure Purcell enhancement of some SPEs?

2. The great care was taken by authors to explain the fabrication process including re-orientation of crystallographic structure Au porous films and growth dynamics by providing sufficient evidence for the claims stated. However, as authors mentioned, de-wetting of Au film was not stopped by the 2D overlayer unlike the work quoted in ref 24. (Coa, P. et al. *Advanced Materials* 29,1701536, (2017)). This makes the outcome of the annealing process appears to be ambiguous which might lead to difficulties in reproducing the results by other scientists. Could authors compare more thoroughly their work with Coa et. al. and elucidate about the reasons why in their case de-wetting is not fully suppressed?

3. The next major concern is applicability of presented results. Authors write in the abstract: "Our results suggest the OPEN film geometry is a versatile platform that could facilitate the use of layered materials in quantum optics systems".

I am afraid I don't see a clear advantage of OPEN against lithographically defined structures in terms of providing "a versatile platform that could facilitate the use of layered materials in quantum optics systems". Creating OPEN by annealing can only follow log-normal distribution, as

described in the main text (ref. 27-29). In contrast to the presented de-wetting mechanism, electron beam lithography allows for much more versatile approach to design a porous metal film (size, distribution of pores). Some advances have been made in this direction from coupling the SPE to a plasmonic waveguide (ref 15). Moreover, there is very little ground covered in the manuscript about the potential impact and benefits of OPEN-2DC for quantum optics systems. Could authors expand on this? Could authors demonstrate that OPEN structure can be created by other means than thermal annealing for example by electron beam lithographically? I believe this would broaden the appeal of the manuscript.

4. A demonstration of a novel device based on OPEN-2DC would be much in place, otherwise I have difficulties to appreciate the novelty of this work which is appropriate for Nat Comms.

My more detailed comments include:

- What is the coupling efficiency between SPP and SPE?
- Is there a correlation between location of pores and SPEs? Does the density of pores correlate with the density of SPE in Fig4a?
- I would appreciate statistics and data supporting the claim that nearly 100% of times 2D crystal is suspended. What is the % of 2DC coating the inside of a pore. It would be nice to see AFM cross sections in Fig.2a. From the AFM scans it seems like the film's height is getting more textured with annealing time. Does the RMS roughness increases with time as the pores are getting bigger?
- The impact on this manuscript depends on the quality of the SPE. Could authors include more detailed characterisation of SPE i.e. linewidth, brightness, spectral jitter?
- What are thicknesses of 2D material in each sample?
- I appreciate the sound of acronym OPEN. However, the full name - orientated pore enabled network, is not intuitive, hard to understand and it describes only a part of the results as it does not include the crucial involvement of 2DC. Other names mentioned in the manuscript are better candidates for example "self-assembled, highly-textured porous metallic framework" mentioned in conclusion or "porous metallic networks".

I recommend improving the readability of figures: i) labels are not clear to read where there are not boxes behind the text (for example Fig.1d or Fig.1i)

Fig.1a the result demonstrates only a single step annealing. In order to include the last graphic (OPEN-Au / 2DC / OPEN-Au / SiO₂) please include the data which exhibits this structure.

Fig.1d Why OPEN has a different contrast than non-capped Au under SEM?

Fig.3c Is this spectrum taken at RT? I suggest adding PL spectrum of WS₂ on SiO₂ for the comparison where there is no quenching due to metal or Au-assisted increase of recombination rate of spontaneous emission at the pores' edges.

Fig.3e It is not clear what is plotted. Integrated or peak intensity? What wavelength range? Scale bar, log or linear scale?

Fig.4 Are PL maps of a and b cover the same area?

Fig. 4a It is not clear what is plotted. Integrated or peak intensity? What wavelength range?

Fig. 4b It is written that the PL map was taken by exciting at the 'dot' location and collecting over entire area including the 'cross' location. What are the excitation and collection locations for g₂ data in fig. 4d?

Fig. 4e To obtain the red spectrum, authors write that they used "excitation at the cross and collection at the dot". However, the opposite is true for the PL map in fig. 4b. If the locations are indeed reversed, what is the purpose of this?

Reviewer #2 (Remarks to the Author):

This work studied the interaction between two-dimensional crystal (2DC) layers and self-assembled porous metallic network. First, the author found that annealing such a Graphene/Au/SiO₂ stack can result in a 'reverse epitaxy' process of the encapsulated Au layer. Next, when using a semiconducting 2DC layer, the PL occurs at the 2DC/metal-pore sites and is quenched at the 2DC/metal regions. The porous metal framework supports propagating surface plasmon-polaritons (SPPs) launched by either a direct laser field or light emitted from the monolayer semiconductor. Finally, at low temperature, the author measured both local and non-local single photon emission, and confirmed the quantum nature of these emissions.

I think this work is interesting, and should be published. Minor revisions are needed, such as:

1. On page 5, before Fig. 2, the in-plane rotations offset is 30 degree, or 60 degree? Different values in main text and in SI.
2. In Fig. 1a, the author showed the structure of 2DC sandwiched by bilayer Au film. I didn't see the discussion of this sample.
3. This is claimed to be a "large-area" exfoliation and transfer of 2DC. So how big the overall area of 2DC is? Are they all single crystal? Did the author use any special method (as in some literatures) to obtain "large-area" 2DC?
4. Unit problem: should be checked over both the main text and the SI.

evidence to support the claim that propagating SPP are responsible for exciting SPEs at different locations. The link between SPPs and SPEs is still unclear to me. Indeed, from NSOM measurements it is clearly shown that a single pore launches a SPP that propagates up to 2 μm . Although it is very important measurement, this does not capture the nature of SPP propagation in the porous film structure.

How does SPP travel through porous metal film?

SPPs will propagate through porous metal films similar to ‘smooth’ metal films but with added losses depending on aspects of the microstructure, such as pore density and pore size. In our samples, SPPs can be excited by two different wavelengths: (i) via the direct laser field through in-coupling at pore sites [²] (532nm here), and (ii) via excitons in the 2D semiconductor [³]. In WSe₂, both neutral (X_o) and charged excitons (X_T) have a temperature-dependent energy that varies from approximately 750nm to 710nm from room temperature down to 4K [⁴]. The Au-induced doping in WSe₂ will result in higher emission intensity from X_T (versus X_o) and therefore, we can reasonably assume the most energetic and intense exciton dipole energy is at approximately 727nm at 4.5K [⁵]. (Note: We have updated our assessment of the WSe₂ exciton energy, which we originally assigned at 750nm, but now are assigning at 727nm, which more realistically reflects the primary exciton population at 4.5K for doped WSe₂).

If SPPs are responsible for propagating energy across the film, the total PL decay should be described by

$$I = I_b + I_{532nm} \frac{1}{x} e^{-\frac{x}{\gamma_{532nm}}} + I_{727nm} \frac{1}{x} e^{-\frac{x}{\gamma_{727nm}}} \quad (1)$$

where λ refers to the exciting wavelength, I_λ is the initial wavelength specific intensity at the launching site, γ_λ is the wavelength specific decay length, and I_b is the background signal. The additional $1/x$ term arises from SPPs radiating in all directions in the 2D plane [³]. To gain insight into the most efficient transmission pathways in the OPEN film, we apply the above equation to the most intense PL data from the single-photon emitter (SPE) from Figure 4f in the main text (Figure R1 below). The most intense PL points represent the highest efficiency pathways for energy in-coupling, SPP propagation, and energy out-coupling. The excellent agreement between Equation 1 and the most intense PL data— over three orders of magnitude— provides high confidence that the PL intensity can be modeled by conventional SPP decay for the remotely excited SPE site in Figure 4 of the main text.

A good fit of Equation (1) occurs with $\gamma_{532nm} = 1\mu\text{m}$ and $\gamma_{727nm} = 15\mu\text{m}$. This implies that the pores do not catastrophically quench SPP propagation. Specifically, the fitted decay length $\gamma_{532nm} = 1\mu\text{m}$ is consistent with decay dominated by Ohmic losses (i.e., pore scattering is not a significant contributor to SPP decay; calculations in revised SI and Fig. R6 below). On the other hand, if one maintains the assumption of only Ohmic losses and examines decay at $\lambda = 727$ nm, analytical treatments predict $\gamma_{727nm} = 32 \mu\text{m}$. Our fitted value ($\gamma_{727nm} = 15 \mu\text{m}$) is 53% smaller, implying that there are additional SPP scattering mechanisms, likely from the pores. This implies the OPEN film structure does contribute additional SPP scattering/loss but that this occurs most significantly for longer wavelengths. The fitted decay length here for $\gamma_{727nm} = 15\mu\text{m}$, by nature, includes all SPP loss mechanisms, (e.g., Ohmic loss, inhomogenities, pore scattering).

Figure R1: Plot of the most intense PL data taken from the remote PL map in Figure 4b, together with a fit from Equation (1) (red line). The individual components of the fit are labeled on the plot, and the total fit (red line) has a $R^2=0.92$, where $I_b=200$, $I_{532nm} = 9.6E5$, $I_{727nm} = 6E3$, $\gamma_{532nm} = 1$, $\gamma_{727nm} = 15$.

The following section shows additional COMSOL simulations to provide a qualitative assessment on how modeled SPP energy varies across a film with and without pores:

We performed 3D COMSOL simulations to obtain a qualitative picture for how modeled SPP decay occurs for a system with similar morphology to the OPEN film structure. Figure R2 shows results from simulations for a 50nm thick Au film on 100nm SiO₂/Si, both with and without pores (model length, width = 15μm, 1.1μm; pore diameter = 230nm). The cross-sectional power dependence from these simulations is shown in the bottom plot for both the 532nm and 727nm excitation. In this plot we show a cross-sectional slice adjacent to the pores (dotted white line), where the decay is mostly monotonically decreasing.

The main conclusions from these simulations are: (i) for 727nm excitation, the SPP decay length decreases by approximately 55% (from $\gamma=20\mu\text{m}$ to $\gamma=9\mu\text{m}$) between ‘no pores’ and ‘pores’ and (ii) for 532nm excitation the decay length is approximately the same between ‘no pores’ and ‘pores’ ($\gamma=0.65\mu\text{m}$). This is consistent with our data and with our hypothesis that SPPs at both $\lambda = 532$ and 727nm are present, are seen in the PL response, and, for the case of $\lambda = 727\text{nm}$, are responsible for generating remote PL over $\sim 10^3 \mu\text{m}$.

Figure R2: 3D COMSOL simulation of a 50nm thick Au film on 100nm SiO₂/Si with and without a line of pores (diameter=230nm, spacing=1.1 μm ; model length = 15 μm , model width = 1.1 μm). The power decay of SPP waves at different wavelengths is shown in the top three images. The SPP is launched from the left side of the image and propagates to the right side of the image. The bottom plot shows the normalized power cross-section (taken off center at 350nm from the edge; dotted white line) for 532nm and 727nm excitations, both with and without pores. We added an exponential decay (dashed lines) to show the approximate decay length (γ) for each model.

The following section provides additional information and context for estimating SPP transmission efficiencies across the pores in the OPEN films.

It is known that SPPs can transmit their energy across gaps, where the transmission probability depends on the gap distance. In gold films, the transmission rate across a 1 μm gap is $\sim 50\%$, for a 200 nm gap is $\sim 80\%$, and for ≤ 30 nm gaps it approaches 100% [6]. For comparison to our system, we note that the OPEN-Au films do not have perfect ‘gaps’ as studied by Flynn et al. [6], but instead have a continuous conducting surface of gold. From this, it is reasonable to assume the transmission values in [6] serve as a LOWER bound for comparing transmission losses of pores to that of true metal gaps.

In Figure 2 of the manuscript we show an example of the pore distributions with annealing a WS₂/Au sample. In this experiment, we measured the ensemble average pore areas of 0.028 μm^2 (resulting in 8.9% pore area coverage) and 0.154 μm^2 (resulting in 29.6% pore area coverage-- discussed in main text), following annealing treatments of 15 minutes and 360 minutes, respectively. Assuming a simple circular hole shape (area= $1/4 * \pi * d^2$), the ensemble average pore diameters (d) in these samples are approximately 190nm and 440nm, respectively. For similar ‘gap’ dimensions from Flynn et al. [6], we could estimate a lower bound for SPP transmission efficiencies at 60-80% across a wide range of OPEN film morphologies.

As labeled in the manuscript and described in more detail here, Figure 4 was acquired using a ‘sandwich’ sample with TWO layers of gold– Au_{25nm}/WSe₂/Au_{25nm} – as illustrated in Figure 1a (bottom) in the main text (also see Figure R3(A) and R9.1 below). The average Au thickness is 50nm and the average pore diameter with exposed WSe₂ in this region is

230nm \pm 88nm (Figure R3(B)), which would correspond to a gap transmission efficiency from Flynn et al. [6] at approximately 80%. In addition, from the pore areal density we can calculate a linear pore density (L_d) of 1.1 pores/ μ m. Taken together, to estimate loss associated with linear scattering between points ‘A’ and ‘B’ due to a pore transmission rate, we assume that every ‘ L_d unit’ the intensity drops by a fixed percentage determined by:

$$I(x) = I_0(T^{x*L_d}) \quad (2)$$

Applying our experimental L_d and variable transmission efficiencies (T), we can compare the loss associated with pore scattering and that associated with the known exponential Ohmic losses in the metal itself (i.e., $I = I_0e^{-x/\gamma}$; note: in the linear approximation, we do not apply the 2D-radial spreading term ($1/x$)). As shown in Figure R3(C) below, a pore transmission rate of T=50% would produce similar loss to that expected from Ohmic loss in gold at $\lambda=532$ nm excitation.

To estimate the total loss due to an Ohmic component and a pore transmission component we use the product of:

$$I(x) = I_0 \left(e^{-\frac{x}{\gamma}} \right) (T^{x*L_d}) \quad (3)$$

In Figure R3(D), we plot the normalized data from the most intense PL in the “WSe₂-exciton” plasmon region since we know that additional loss from the porous metal film occurs at this wavelength. A good overlap of Equation (3) occurs with a pore transmission rate of approximately 94%. As reasonably expected, the best transmission rate for the average 230nm pores is larger than that measured for a 230nm gap assessed by Flynn et al. [6], where we estimate the ‘pore transmission’ here to be about 18% larger than the ‘gap transmission’ in Flynn et al. [6]. Finally, we reemphasize that this assessment characterizes the most efficient combined process for exciton excitation (in-coupling), SPP excitation/propagation, and exciton re-excitation (out-coupling). From the plot in Figure 4f of the main text, intensity points that fall below these most intense PL data represent less efficient exciton/SPP propagation/exciton pathways in the OPEN film and shorter propagation lengths.

Figure R3: (A) AFM from the region in Figure 4 of the main text showing positions of the central collection spot (labeled 'collection'), and the position of the excitation spot at $17\mu\text{m}$ distance (labeled 'excitation'). (B) Particle counting via ImageJ software to produce a mask of the pores where WSe_2 is exposed. (C) Plot comparing loss due exponential decay (solid line, $I=I_0 e^{-x/\gamma}$) and that for fixed transmission loss (dotted line) from Equation 2. (D) Plot showing normalized data from the WSe_2 -exciton plasmon region, together with Equation (3) using different transmission values.

How such short-travelling SPP can excite SPEs that are $17\mu\text{m}$ away from the laser spot?

As the Reviewer noted, we used NSOM measurements to directly measure SPP waves launched by 532nm excitation at a pore site. As demonstrated in Ref [13] from main text, it is also possible to launch SPPs via excitons in a capping TMD layer. Rather than emitting from the film vertically as PL, the energy released upon recombination of the exciton is coupled into a SPP mode. Coupling energy from the exciton to the SPP mode is analogous to a dipole on the surface of the metal, which can meet the necessary wavevector conditions to couple light into an SPP mode. Therefore, in our OPEN-TMD films we observe BOTH SPP modes, one directly excited by the incident laser at the pore sites (532 nm here), as well as those excited through exciton recombination in the TMD layer (727nm in WSe_2) at the pore sites.

The total intensity of a propagating SPP wave in a metal film is a combination of its intrinsic decay length (γ) and the power applied to excite the wave (Equation 1). Equally important, γ is also wavelength dependent. The excellent fit of Equation 1 demonstrates that with a decay length of $\gamma=1\mu\text{m}$, intensity from the 532nm SPP is still measurable up to $\sim 6\mu\text{m}$ away. However, the intensity decreases by 3-orders-of-magnitude as expected. As we have shown above, the PL intensity beyond $\sim 6\mu\text{m}$ becomes well described by a decay constant of $\gamma=15\mu\text{m}$. It is this longer decay constant that we attribute to the expected 'exciton-excited' SPP mode, which is at a longer wavelength (727nm) than the excitation laser, and is able to propagate at least $17\mu\text{m}$ to remotely excite the SPE under study.

Would cross-correlation ($g_2(\tau)$) measurement between two different ‘non local’ SPE provide an insight to involvement of SPP? A SPE that is further from the origin of SPP should in principle emit later than a SPE that is closer?

If we understand the question correctly, given the fast propagation time of light (e.g., to propagate 17 μm is: $17 \mu\text{m}/c = 5.67 \times 10^{-14}\text{s}$), it is not in our measurement capability to differentiate between different emitters sites excited at different distances by SPPs.

As demonstrated in Zhou Y. et. al. Nature Nanotechnology 12, 856 (2017), SPP couple to dark excitons in WSe₂. Do authors observe such enhancement of PL emission from dark excitons in monolayer WSe₂ enabled by OPEN?

The work by Y. Zhou et al. [5] makes use of a dielectric layer to isolate the WSe₂ from the Ag plasmonic waveguide which doubles as a gate electrode. In this geometry, it is possible to change the doping of WSe₂ between, *n*-type, intrinsic (*i*), and *p*-type. In contrast, the TMD layers in the OPEN-films are in direct contact with the supporting Au substrate and doped as a result. As shown in Figure 2e of the main text, direct connection between the TMD metal induces surface potential fluctuations across the film and variable doping. Therefore, while the prospect for coupling to dark excitons remains, it is currently not possible to independently modulate the TMD doping to explore this mechanism.

To provide further insight into this question, we have also performed additional COMCOL simulations to examine how well the *in-plane* (e.g., neutral excitons) and *out-of-plane* (e.g., dark excitons) dipoles couple into SPP modes of the metal, similar to Zhou Y. et al. [5]. The most important result from these simulations is that the *in-plane* dipoles couple into SPP modes by orders-of-magnitude larger when a pore is present in the metal, as compared to a continuous metal film as studied in [5]. The resulting $|E_z|^2$ intensity is close to three (3) orders-of-magnitude more intense in the presence of a pore (Figures R4 and R5). This result also highlights that propagating SPP waves will couple much more strongly into dipoles located within pores.

Figure R4: 2D COMSOL simulations showing the field intensity component E_y from an oscillating dipole (‘red dot’ in schematic), both in-plane and out-of-plane, above a gold surface with and without a pore. The dipole position is labeled relative to the dotted line at the Au surface and is either 10nm above the Au surface ($z=10\text{nm}$) or at the bottom of the pore ($z = -49.5\text{nm}$) just above the SiO₂ surface. The simulation length is 10 μm with the dipole centered at $x=5\mu\text{m}$. The material thicknesses are all the same as labeled in the top schematic (note: schematic not to scale). The pore diameter is 230nm.

Figure R5: Intensity cross-section taken from the 2D COMSOL simulations in Figure R4 at 50nm above the surface, where the dipole is centered at $x=5\mu\text{m}$. The plot shows the square of the field intensity ($|E|^2$). The presence of the pore results in significant improvement for the in-plane dipole coupling, where $|E|^2$ is close to three orders-of-magnitude larger than a continuous film (left plot) away from the emitter site.

Authors use work mentioned in ref. 13 and 36 to support their interpretation of the data. In ref. 13, it has been demonstrated that SPP can propagate up to $34\mu\text{m}$. This was achieved thanks to a smooth interface of 2D metal-oxide-semiconductor (Ag and WS₂). On the other hand, in this work porous morphology annealed metal films provides many out-coupling sites for SPP due to numerous terminations of metal layer which effectively create sharp dielectric changes between air and metal. From ref. 36: “The attenuation length is affected by the scattering of SPs on the surface roughness, scratches, and inhomogeneities of the metal film or the adjacent medium [therein 15,16]”. Intuitively, this should decrease SPP propagation. Having this in mind I believe that it is unlikely that SPPs propagating along a porous metal are responsible for exciting SPEs at distances on the order of 10s of μm .

As the Reviewer notes, in Ref. [13] of main text, SPPs were estimated to propagate up to $34\mu\text{m}$ in a ‘smooth’ sample consisting of TMD/ Al_2O_3 /Ag layers. To help readers more quickly compare propagation lengths in different materials, we have added a plot in the SI of calculated propagation lengths for Ag and Au using physical constants from [7]. We also highlight another important difference between our 2DC/metal interface and that in Ref [13] of main text. In Ref [13] they deposit an amorphous oxide on their metal surface, which does not produce a highly-crystalline, well-defined interface. In our OPEN-2DC samples, the ‘reverse epitaxy’ process produces a perfectly crystalline, atomically aligned and atomically smooth interface (Fig R7, R8 and discussion below). The OPEN-2DC film has atomically flat terraces that are separated by metal atomic step bunches. Such an interface should have lower atomic potential fluxuations as compared to an amorphous oxide/metal interface.

Figure R6: Calculated SPP propagation lengths for Ag and Au using optical constants from [7]. The red stars indicate the longest propagation lengths measured in Figure 4 of the main text. Labeled “Ref. 13” and “Ref. 36” are from the main text.

We also agree with the Reviewers interpretation that morphological features of a thin metal film will decrease SPP propagation, where specific features were discussed and analyzed in Ref. [36] of main text. To be more realistic for our system that is why we referenced the assessment of Ref. [36] of main text to help understand reduced SPP propagation due to morphological features in the metal film. Without morphological feature scattering, $\gamma_{727\text{nm}} = 32\mu\text{m}$ in Au versus our fitted value of $\gamma_{727\text{nm}} = 15\mu\text{m}$. As discussed above, our incorporation of the experimental results by Flynn et al. [6] demonstrate that the gaps in our metal films do not add a significant amount of scattering, despite the presence of edges and a physical break in the metal.

The Reviewer makes a final important point here in that they believe it is unlikely that SPPs can travel on the order of 10s of μm in these porous metal films. We stress that it is this ‘unexpected’ nature of our results that makes it significant and that our demonstration of both the ‘reverse epitaxial’ process, together with non-spatially local SPE emission, that makes our manuscript suitably novel for Nature Communications. The additional analysis provided above provides confidence that SPPs can travel on the order of 10s of μm in these porous metal films.

There is an ambiguity in estimating SPP propagation length. On one hand, authors use the work from ref. 36 to give the length of 15 μm for 750 nm plasmons, where the influence of metal roughness is studied. On the other, authors use a formula for Ohmic losses (ref), (mentioned in the section 8 in the supplementary) which assumes flat and smooth interface to obtain 1 μm propagation length for 532 nm plasmons. In **Fig.4f** Plotted lines for gamma do not reflect well the data. Are three lines fitted values or fixed values? What are the fitted values for decays? *Could authors improve their strategy for estimating, measuring and fitting SPP propagation length?*

Based on the Reviewers recommendation, we have performed a more rigorous analysis of the PL intensity decay as shown and discussed in Figures R1, R2, and R3 above. Decay length values of $\gamma = 1\mu\text{m}$ and $\gamma = 15\mu\text{m}$ produce a strong statistical fit, consistent with the expected decay lengths for plasmons excited by green or NIR light in this material.

We thank the Reviewer for pointing out a potential source of confusion when discussing both the Ohmic loss equation and the assessment discussed in Ref [36]. We state in the manuscript “The propagation length (γ) of an SPP in our system, dominated by resistive losses in the metal,

is estimated at approximately $\gamma \approx 1 \mu\text{m}$ when excited by a 532 nm laser field. [33,36]”. We cited BOTH ref. [36] and the ‘Ohmic loss’ equation.

To clarify this point, we note that at shorter wavelength excitations ($< \sim 600\text{nm}$), the SPP propagation length for Au calculated from ref. [36] and that calculated from the Ohmic loss equation begin to converge. This is due to the fact that at lower excitation wavelengths the impact of morphological scattering becomes *less than* that due to Ohmic losses in the metal itself. We also see this trend in our COMSOL simulations (Figure R2), where the decay length for a model ‘with pores’ and ‘without pores’ is approximately the same at 532nm excitation, but different at 727nm excitation.

As the skin depth of SPP is larger than the thickness of metal film used, SPP propagation along OPEN would be modulated by the roughness proportional to the given depth of sharply terminated pores (25 nm) and their density. *How the roughness of OPEN film is taken into account to estimate SPP propagation length?*

By applying Equation 1 to the most intense PL data (Figure R1), we extract the longest propagation length γ for the OPEN film. This γ value includes in it ALL loss mechanisms in the OPEN film, which includes Ohmic loss, roughness, pore transmission, inhomogeneties, etc... We do not need to know *a priori* what the individual loss mechanisms are to measure the propagation length in the film. To gain quantitative insight into what the additional SPP loss might be over that of Ohmic loss due to pore-to-pore transmission, we preformed the assessment in Figure R3.

For the single-photon emitter studied in Figure 4, we used a multi-layer ‘sandwich’ OPEN metal structure (Au/WSe₂/Au), where the starting Au thickness of each layer was approximately 25nm. The average metal thickness across the sample is approximately 50nm.

To more quantitatively address the Reviewers questions regarding surface roughness, we have included additional high-resolution AFM and STM scans of an OPEN-WS₂ film (Fig. R7 and R8 below). By doing so, we find our 2DC-passivated gold surfaces can be atomically flat for any given Au grain/terrace, with the possibility of single Au steps or step bunching. The measured RMS roughness at the tens-of-nanometer scale (via STM) is $\sim 10\text{pm}$, where we always observe the expected moiré lattice between the TMD and Au (i.e., perfect crystalline interface). From AFM, the rms roughness (R_a) of a single Au terrace (at the micron scale) measure approximately 0.15nm and across multiple Au terraces measures approximately 1nm. This assessment of TMD/Au roughness suggests there should be limited SPP scattering due to the type of surface roughness examined in Ref [36], which did not have atomically flat Au crystallites.

Figure R7: Higher resolution AFM images of a WS₂/Au sample annealed at 300°C. The surface roughness is analyzed

for different regions, ranging from individual Au terraces (a) to multi-terrace areas (c). (b) Example of height variation across the surface due to step bunching of monoatomic Au layers.

Figure R8: STM images were taken in a ScientaOmicron LT STM under UHV ($\sim 10^{-11}$ Torr) conditions with LN₂ cryostat cooling to 78K. Constant-current feedback was used for imaging the surface with an electrochemically etched tungsten tip which was gold plated on sputter-cleaned Au(111) on mica. The sample was aligned under the STM tip by an optical window in the thermal shielding and long-range ex situ microscope. Resulting data was analyzed with WSxM for assignment of RMS roughness and lattice highlighting. In both (a) and (c) the moiré pattern formed between MoS₂/Au is clearly observed, indicating an atomically clean interface.

The SPPs propagate in directions determined by the polarization of the incident beam. If the claims in the manuscript are to be accurate, I would expect that different polarisations should yield to different intensity distribution of PL maps that are fostered by OPEN. *So, why the PL map in fig. 4b does not reflect this property of SPP and instead the PL map seems to have a circular symmetry which one would expect to observe when exciting with a circularly polarised light?*

We agree with the reviewer that SPPs are influenced by the polarization of the incident beam and during our studies we have examined if there is a transferable effect on the resulting PL intensity. As shown and discussed in the text, our NSOM measurements do show the expected polarization dependence for SPPs excited by the incident 532nm beam. However, we have not yet observed differences in the PL intensity or directionality based on polarization information being retained by the SPP-exciton excitation process.

Upon further consideration this is perhaps not a surprising result. There are two pathways in which SPPs are excited in our samples: (i) by the polarized laser field via pores in the metal (532nm excitation) and (ii) by exciton dipoles in the 2D semiconductor. Regarding pathway (ii), polarization information does not appear to be retained in the exciton-SPP excitation process, which was also observed in Ref [13] of main text (see Fig. 2a in Ref. [13]). In addition, our COMSOL simulations show that the coupling intensity of both in-plane and out-of-plane dipoles into the porous metal structure are on the same order of magnitude (Fig. R4, R5).

This makes me think that there might be another mechanism that enables the dispersed excitation of SPEs. An alternative effect could be that the metal surface with its numerous holes, sharp edges and variations in height scatters efficiently excitation light in all directions which could optically and directly excite neighbouring SPEs. This seems plausible to me since the sufficient excitation power for observing SPEs is in the nW range,

as shown in fig. 4e. Moreover, to acquire the PL map shown in fig.3e, 4.5mW of pump power was used which is remarkably high amount of power, suggesting that the scattering might play a role. *Could authors attempt to disprove or confirm this hypothesis?*

We agree with the Reviewer that it is important to provide a control experiment to test for, and ultimately exclude, other possible excitation mechanisms.

First, it is important to clarify that both the experimental setup and the TMD materials in Figure 3 and Figure 4 are different. In Figure 3e we collect a fluorescence image (CCD image) of a WS₂ sample that is illuminated with a laser spot at room temperature (4.5mW). It is not a spatial PL map formed by scanning an excitation or collection spot across the surface with a spectrometer/detector. The data in Figure 3e is presented with a truncated LINEAR intensity scale to highlight the weaker fluorescence/PL that occurs away from the excitation spot. The data in Figure 4a was acquired using conventional PL mapping (at 4.5K) of WSe₂ where the laser excitation (10μW) and collection spot are confocal. In contrast, the data in Figure 4b is obtained by fixing the collection aperture at the center spot (black dot) and scanning the excitation across the surface; intensity is plotted on a LOG scale. In Figure 4 the maximum excitation power used was 10μW.

Regarding the possibility of scattered 532nm light directly exciting a SPE emitter:

(Point 1) The sample studied in Figure 4 is a ‘sandwich’ sample of Au/WSe₂/Au, such that the semiconducting layer is not in-line with the top surface. This geometry will significantly reduce the probability that a 532nm scattering event will directly excite the buried semiconductor at a different location. Figure R9.1 shows an example of a high-resolution AFM image of another Au/WSe₂/Au sample demonstrating how the WSe₂ layer is located below the top surface, thereby further protecting it from low-angle scattered light.

Figure R9.1: (A) AFM height image of a ‘sandwich’ Au/WSe₂/Au sample. (B) Cross-section profile from the red line in (A). (C) Schematic showing the location of the different layers in the structure.

(Point 2) We have performed an additional control experiment at the same region mapped in Figure 4 to examine the extent of scattered 532nm light from the OPEN film structure. In this control, we perform the same measurement as shown in Figure 4b, but use a 532 nm band-pass filter to only collect scattered 532 nm light (and not PL light). In Figure R9.2 below, we observe that the intensity of scattered 532 nm light drops off at a greater rate than the PL intensity starting at approximately 0.5 μm away from the excitation spot. By 3 μm from the excitation spot, the 532nm signal intensity is at the baseline intensity of ~200counts/s.

Figure R9.2: Three different examples of strategies to lithographically define where dewetting occurs, and subsequently where PL emission from pore sites occurs.

If it is true that SPPs excite SPEs at neighbouring pores sites, I would expect that the intensity distribution of PL maps from fig.3e and fig.4b should look similar. However, these PL maps appear to look very different. *Why is that? How would a histogram of peak intensity vs distance (as shown in Fig. 4f) would look like for PL map from Fig. 3e? Why not integrated intensity?*

As mentioned earlier, the data from Figure 3e and Figure 4b were collected in different experimental setups using different TMDs (WS₂ versus WSe₂), and are plotted on different scales (truncated LINEAR, versus LOG). In Figure 3e we collect a CCD image (fluorescence image) of a WS₂ sample that is illuminated with a laser spot at room temperature. The data is presented on truncated linear scale so that the weak emission far away is highlighted, while bright emission is saturated on the image. In Figure 4b, the map is formed on a sample consisting of WSe₂ at low temperature (4K) by a scanning excitation beam across the surface with an excitation power of 10μW. The intensity in Figure 4b is plotted on a LOG scale and none of the data points are saturated on the image.

In such systems like OPEN-WSe2 one could expect Purcell enhancement to be relevant and pronounced in SPE. *Do authors observe such effects? Is this surprising that the ‘non-local’ emitter from fig. 4b,d,e shows even longer decay time than ‘local’ emitter? Can authors measure Purcell enhancement of some SPEs?*

We thank the reviewer for raising this interesting possibility and agree that the OPEN films are an attractive platform for additional studies on the quantum optics of SPEs in a richly textured, nanoplasmonic environment. The sharp nanoscale metal features and suspended 2D crystal drums present in OPEN materials may be useful in modifying the local field strength / local density of states at the site of a SPE. We are actively pursuing further experiments to look for Purcell enhancement or other relevant light-matter interactions enabled by this new material, and we suspect other researchers may wish to do the same. We have added language to the manuscript to point out the relevance of OPEN films for these potential applications.

In terms of the specific question about the lifetimes we present in this paper, we expect that the observed lifetime (extracted from photon correlation measurements) will depend both on the local density of states present at the emitter location (i.e., Purcell effect) and the detailed photophysics / level structure of the emitter itself. Experimentally, we find that the lifetime of some emitters depends in a complicated way on optical pump power, and we do not expect that the effective pump power will be the same between local excitation at 25 nW and remote excitation from 10 uW at an in-coupling site 17 um away. The difference in lifetimes apparent from the data in Fig 4c,d may be related to the local/remote excitation mechanism, and it may also scale with the effective power reaching the emitter in each case.

2. The great care was taken by authors to explain the fabrication process including re-orientation of crystallographic structure Au porous films and growth dynamics by providing sufficient evidence for the claims stated. However, as authors mentioned, dewetting of Au film was not stopped by the 2D overlayer unlike the work quoted in ref 24. (Coa, P. et al. *Advanced Materials* 29,1701536, (2017)). This makes the outcome of the annealing process appears to be ambiguous which might lead to difficulties in reproducing the results by other scientists. *Could authors compare more thoroughly their work with Coa et. al. and elucidate about the reasons why in their case de-wetting is not fully suppressed?*

There exist experimental differences between our work at that of Cao et al. One important difference is the use of hydrogen gas during the annealing process, which we explicitly address in the main text. As stated in the main text: “The Au film properties, substrate and annealing conditions were selected to promote metal dewetting at low temperatures even in the absence of the 2DC layer. We find the addition of hydrogen gas to the annealing ambient reduces the onset temperature of dewetting by at least 100°C as compared to Ar alone (Fig. 1b,c), presumably through the reduction of suboxides and/or local carbonaceous pinning sites.” In Cao et al., they perform their experiments using a nitrogen-only ambient. We specifically show in Figure 1b,c the difference in using hydrogen gas during annealing for a Au/SiO₂ sample. In that example, at 300C under Ar-only ambient there is NO dewetting of Au/SiO₂, while at 300C under H₂/Ar ambient there is significant de-wetting.

We show in Figure 2d how the presence of defects in the 2DC layer also impacts the dewetting process. In general, CVD grown layers such as that grown by Cao et al. have more defects than exfoliated layers and the de-wetting results can vary more from sample-to-sample using CVD films depending on the material quality. In Figure 2d we highlight in a dramatic fashion how the underlying Au dewetting process can be moderated from defects in the capping 2DC. There likely exist differences in the crystalline quality between the CVD-grown graphene films in Cao et al., and the exfoliated materials used here (e.g., polycrystalline versus single-crystal, wrinkle and defect density, etc...), which can lead to decreased dewetting as observed in Cao et al.

3. The next major concern is applicability of presented results. Authors write in the abstract: “Our results suggest the OPEN film geometry is a versatile platform that could facilitate the use of layered materials in quantum optics systems”.

I am afraid I don't see a clear advantage of OPEN against lithographically defined structures in terms of providing “a versatile platform that could facilitate the use of layered materials in quantum optics systems”. Creating OPEN by annealing can only follow log-normal distribution, as described in the main text (ref. 27-29). In contrast to the presented dewetting mechanism, electron beam lithography allows for much more versatile approach to

design a porous metal film (size, distribution of pores). Some advances have been made in this direction from coupling the SPE to a plasmonic waveguide (ref 15). Moreover, there is very little ground covered in the manuscript about the potential impact and benefits of OPEN-2DC for quantum optics systems. *Could authors expand on this? Could authors demonstrate that OPEN structure can be created by other means than thermal annealing for example by electron beam lithographically?* I believe this would broaden the appeal of the manuscript.

Our goal in writing the final line of the abstract was to motivate the work in the broader context of developing tools and techniques that will advance capabilities to link quantum emitter sites across a surface. As such, we have modified this sentence to read: “Our results suggest design criteria for metal/2DC systems that could facilitate the use of layered materials in a wide variety of systems involving quantum emitters and plasmonic nanostructures.”

To our knowledge, this is this first demonstration of exciting a quantum emitter site remotely at a distance of $17\mu\text{m}$ in a 2DC/metal system. In this context, what our work does is demonstrate important design criteria that, to our knowledge, have not been discussed in the literature. We believe the key morphological aspects of our process that help enable this long-distance excitation are: (i) the crystallographic alignment of the 2D semiconductor and underlying metal film and resulting perfect crystalline interface, (ii) pores in the metal that allow for coupling between SPPs and exciton dipole emitters in the 2D semiconductor, (iii) avoidance of multi-step lithographic processes and their potential for contamination or material degradation from exposure to resists, wet chemicals, etching processes, etc.

As we have shown in the main text, there exist several strategies to pattern where pores form during the OPEN-2DC film processing. One strategy is to define ‘pinning’ sites for the metal layer so that it cannot de-wet during thermal annealing (Figure 3d main text). Alternatively, it is possible to lithographically modify the 2DC capping layer to define pore formation (Fig. 2d main text). As a third example, we show here (Figure R10 below) the possibility of lithographically etching the substrate to direct where pore forms and hence, where emission sites are located. In this example, we form linear arrays of emitter sites in an OPEN film with a *long-term* goal of incorporating wave-guiding functionality (which is beyond the current scope of work).

Figure R10: Three different examples of strategies to lithographically define where dewetting occurs, and subsequently where PL emission from pore sites occurs.

4. A demonstration of a novel device based on OPEN-2DC would be much in place, otherwise I have difficulties to appreciate the novelty of this work which is appropriate for Nat Comms.

We agree that the specific demonstration of a novel device is an important long-term goal for the new discoveries reported in our work. To show a pathway forward, we have provided three (3) different examples of how the OPEN film, and subsequent luminesce, can be lithography patterned using different techniques: (i) patterning adhesion layers beneath the metal (Fig. 3d), (ii) introducing defects into the 2DC layer itself (Fig 2d), (iii) preforming dewetting on textured substrates (Figure R10).

For this manuscript, we argue that a new device structure is unnecessary for suitability in Nature Communications given the level of detailed required to describe these new discoveries. We further stress that in the Reviewers own opinion, they believe that it is unlikely that SPP propagate 10s of microns in these samples. We have provided significant additional information to give confidence in this interpretation. This supports the novelty of the work and is an important reason for publication in Nature Communications. We have demonstrated a new way to form hybrid 2DC/metal systems that support propagating SPPs and allow for in-coupling and out-coupling of energy via excitons in the 2D semiconductors.

My more detailed comments include:

- What is the coupling efficiency between SPP and SPE?

To help provide a qualitative picture for how SPP energy is entering a pore site, we examine cross-sectional profiles from an individual pore shown in Figure R3. We selected a pore that is 10.5 μm from the launching site to verify that a measurable amount of SPP energy is still available to interact with excitons in the 2D semiconductor. The primary take away from Figure R11 below is that while the SPP power within the continuous metal film is small, significant SPP power does exist within the pore, where there are intensity ‘hot spots’ both at the top surface (Au/Air) and at the bottom surface (Au/SiO₂) and thereby can interact with a 2D semiconductor that is either suspended above or coating the inside of pore. As discussed in Figure R4 and R5 here, an *in-plane* dipole found within a pore (either at the top or bottom of the pore) can much more efficiently couple into SPP modes in the metal as compared to the geometry of a planar metal surface. We surmise that it is the unique geometry of a pore with embedded 2D semiconductor that facilitates the coupling between SPP and SPEs.

Figure R11: 3D COMSOL simulation showing the 727nm SPP power across an individual pore (diameter = 230nm). This data is taken from the tenth pore shown in Figure R2 and is 10.5 μm from the launch site.

- Is there a correlation between location of pores and SPEs? Does the density of pores

correlate with the density of SPE in Fig4a?

Yes. As shown schematically in Figure 3a and 3b of the main text, the brightest emission comes from regions where the TMD layer is not directly supported by an underlying gold layer. In the OPEN film geometry this occurs at pore sites. The PL intensity from pore sites is orders-of-magnitude brighter than from the Au-supported regions (e.g., Figure 3c main text). Figure R12 below provides additional qualitative analysis comparing the PL map from Figure 4a with the pore locations identified by AFM. The density of narrow emitters is on the same order as the density of pore sites.

Figure R12: Comparison of PL from a region in Figure 4a (left side) with the identified pores from AFM analysis (middle image). (Right image) Composite image with semi-transparent AFM height image (from Figure R3A) overlaid with the PL map. The absence of PL/pores in the bottom right of the image is due to a hole in the WSe₂ layer. Due to the absence of well-defined fiduciary marks, the alignment within (+/-) 1-2 microns.

- I would appreciate statistics and data supporting the claim that nearly 100% of times 2D crystal is suspended. What is the % of 2DC coating the inside of a pore. It would be nice to see AFM cross sections in Fig.2a. From the AFM scans it seems like the film's height is getting more textured with annealing time. Does the RMS roughness increases with time as the pores are getting bigger?

Per the Reviewers request, we have provided additional data to support this claim. In this example, we use different microscopies to span from low resolution (>100 μ m) to high resolution (5 μ m). In Figure R13 below we increasingly provide higher magnification images from one region on an OPEN-WS₂ film. Using AFM imaging at 30x30 μ m² area, we can begin to differential between suspended membranes and membranes coating the pores. From this image, we can identify nine (9) non-suspended membranes out of approximately 2,500 pores.

Figure R13: Example showing the formation of thousands of pores in an OPEN-WS₂ films. From the 30x30μm² AFM image, we can identify nine (9) WS₂ membranes which coat the inside of the Au pores (red circles).

Per the Reviewers request, we have provided additional data for AFM cross sections in Figure 2a (Figure R14 below).

Figure R14: AFM cross sections taken from Figure 2a of the main text.

Regarding sample roughness, in the images shown in Figure 2a (and Fig. R14 here), the total image roughness factor (R_a) increases with anneal time by ~ 0.6 nm. The measured image roughness (R_a) for 15 minutes: $R_a=3.63$ nm; 60min: $R_a=3.96$ nm; and 360min: $R_a=4.20$ nm. We note that this R_a analysis also includes height variations caused by the slightly depressed WS₂ membranes, in addition to height variations in the Au layer. As a result, these ‘image R_a ’ values are likely higher than that from the Au film itself. As shown in Figure R4 and R5, the individual Au terraces can be atomically flat with step bunching occurring across the surface as metal re-distributes to form pores. Without including pores in the R_a analysis (Fig R4) R_a is closer to 1nm over a few micron area. As seen in the 360min anneal in Figure R14 (Figure 2a main text), the Au grains and terraces increase in size with annealing time, and therefore have fewer step boundaries across the crystalline surface.

- The impact on this manuscript depends on the quality of the SPE. Could authors include more detailed characterisation of SPE i.e. linewidth, brightness, spectral

jitter?

Per the Reviewers request, we have included additional plots in the SI to show more details of the narrow emitters. Figure R15 below shows the region from Figure 4 of the main text, with each individual narrow emission site identified with a red ‘dot’ (Fig. R15(A)). We also show histograms of the emitter peak frequency and brightness, along with representative examples of individual emitters from across the area studied.

Figure R15: Selected narrow emitters from Figure 4a of the main text.

- What are thicknesses of 2D material in each sample?

For most of the annealing studies and spectroscopy results discussed and analyzed here (Figure 2, 3, 4 of main text), we focus on monolayers since, in the case of TMDs, they are direct gap semiconductors and have strong PL. For the image shown in Figure 1d, we present data on a few-layer graphene region. While the exfoliated samples have a wide range in flake thickness, we focus on monolayers to simplify the analysis and discussion.

- I appreciate the sound of acronym OPEN. However, the full name - orientated pore enabled network, is not intuitive, hard to understand and it describes only a part of the results as it does not include the crucial involvement of 2DC. Other names mentioned in the manuscript are better candidates for example “self-assembled, highly-textured porous metallic framework” mentioned in conclusion or “porous metallic networks”.

We appreciate the feedback on the OPEN acronym. We have modified the acronym slightly from ‘Orientated Pore Enabled Network’ (OPEN) to ‘Oriented Porous mEtallic Network’ (OPEN) as suggested. In order to differentiate which 2DC layer is responsible for the ‘reverse epitaxial’ process, we add the specific 2DC name to the name (i.e., OPEN-2DC). For example, if we use WS_2 to re-crystallize Au to form a porous metallic network, we write

'OPEN-WS₂'.

I recommend improving the readability of figures: i) labels are not clear to read where there are not boxes behind the text (for example Fig.1d or Fig.1i)

As suggested, we have added a box behind the figure labels in Figure 1.

Fig.1a the result demonstrates only a single step annealing. In order to include the last graphic (OPEN-Au / 2DC / OPEN-Au / SiO₂) please include the data which exhibits this structure.

As suggested, we have updated the supplemental information to include an AFM image of the Au/ WS₂/Au / SiO₂ sample studied in Figure 4 of the main text.

Fig.1d Why OPEN has a different contrast than non-capped Au under SEM?

In Fig. 1d, the central OPEN region has a different contrast due to the conducting graphene layer being suspended above the SiO₂ layer within an individual pore. Outside of this region, the localized Au islands are on insulating SiO₂ and the electron beam charges the SiO₂ differently in the absence of the graphene layer.

Fig.3c Is this spectrum taken at RT? I suggest adding PL spectrum of WS₂ on SiO₂ for the comparison where there is no quenching due to metal or Au-assisted increase of recombination rate of spontaneous emission at the pores' edges.

All data take in Figure 3 was collected at room temperature. Per the Reviewer's suggestion, we included a new SI Figure comparing WS₂ on the dielectric PMMA (PMMA will have less scattering from substrate charges as compared to SiO₂). There is about a factor of three (3) difference between the OPEN-WS₂ and WS₂/PMMA PL peak intensity.

Figure R16: Comparison of room temperature PL of WS₂ on different substrates.

Fig.3e It is not clear what is plotted. Integrated or peak intensity? What wavelength range? Scale bar, log or linear scale?

Figure 3e is a fluorescence image acquired using a CCD imaging system. It is not a scanned PL map as shown in Figure 4a,b. A notch filter was used to block the 532nm excitation beam and a long pass filter (>600nm) allows fluorescence from the WS₂ layer to reach the CCD. The integrated intensity is shown on a truncated linear scale.

Fig.4 Are PL maps of a and b cover the same area?

Yes, they measure the same area.

Fig. 4a It is not clear what is plotted. Integrated or peak intensity? What wavelength range?

Fig 4a shows the total photoluminescence intensity from each point on the sample as measured by a Si avalanche photodiode placed after a 600 nm long-pass filter. The wavelength range is defined by the 600 nm filter and the detector's sensitivity cutoff in the NIR. The resulting range is approx. 600 – 1000 nm. We have modified the figure to more clearly identify what was measured.

Fig. 4b It is written that the PL map was taken by exciting at the 'dot' location and collecting over entire area including the 'cross' location. What are the excitation and collection locations for g2 data in fig. 4d?

As originally described in the caption, in Figure 4b the collection spot is FIXED at the 'dot' location and the excitation spot is SCANNED to form the image. For Figure 4c, both excitation and collection were at the 'dot' point, while in Figure 4d the COLLECTION was at the 'dot' point and EXCITATION at the 'cross' point.

Fig. 4e To obtain the red 'non-local' spectrum, authors write that they used "excitation at the cross and collection at the dot". However, the opposite is true for the PL map in fig. 4b. If the locations are indeed reversed, what is the purpose of this?

The image in Figure 4b is collected as a 'non-local' map, where the COLLECTION spot is held fixed at the 'dot' point, and the EXCITATION beam is scanned over the area.

REFERENCES

- 1 Desai, S. B. *et al.* Gold-Mediated Exfoliation of Ultralarge Optoelectronically-Perfect Monolayers. *Advanced Materials* **28**, 4053-4058, doi:10.1002/adma.201506171 (2016).
- 2 Yin, L. *et al.* Surface plasmons at single nanoholes in Au films. *Applied Physics Letters* **85**, 467-469, doi:10.1063/1.1773362 (2004).
- 3 Shi, J. *et al.* Cascaded exciton energy transfer in a monolayer semiconductor lateral heterostructure assisted by surface plasmon polariton. *Nature Communications* **8**, 35, doi:10.1038/s41467-017-00048-y (2017).
- 4 Wang, G. *et al.* Valley dynamics probed through charged and neutral exciton emission in monolayer WSe_2 . *Physical Review B* **90**, 075413, doi:10.1103/PhysRevB.90.075413 (2014).
- 5 Zhou, Y. *et al.* Probing dark excitons in atomically thin semiconductors via near-field coupling to surface plasmon polaritons. *Nature Nanotechnology* **12**, 856, doi:10.1038/nnano.2017.106 <https://www.nature.com/articles/nnano.2017.106#supplementary-information> (2017).
- 6 Flynn, R. A. *et al.* Transmission efficiency of surface plasmon polaritons across gaps in gold waveguides. *Applied Physics Letters* **96**, 111101, doi:10.1063/1.3360202 (2010).
- 7 Johnson, P. B. & Christy, R. W. Optical Constants of the Noble Metals. *Physical Review B* **6**, 4370-4379, doi:10.1103/PhysRevB.6.4370 (1972).

REVIEWERS' COMMENTS:

Reviewer #1 (Remarks to the Author):

Dear Editor and Authors,

In the main, I appreciate authors' thoroughness and care with which the questions were answered. The new version of the manuscript gained a lot of clarity. I am mostly but still partly satisfied with the explanations and answers included in the response. There are still gaps in the main text which make the manuscript unclear in some places. In particular, I ask for more clarification in the main text with regards to different decay rates.

I recommend that after attending further suggestions I would recommend the manuscript for the publication in the Nature Communications.

Please find the second set of point by point feedback to authors response and further comments and suggestions in the the attached files.

With Regards,
Reviewer no.1
Art Branny

evidence to support the claim that propagating SPP are responsible for exciting SPEs at different locations. The link between SPPs and SPEs is still unclear to me. Indeed, from NSOM measurements it is clearly shown that a single pore launches a SPP that propagates up to 2 μm . Although it is very important measurement, this does not capture the nature of SPP propagation in the porous film structure.

How does SPP travel through porous metal film?

General comment about Methods:

A. Please include the information where the 2D bulk crystals were supplied from.

Section 1

SPPs will propagate through porous metal films similar to ‘smooth’ metal films but with added losses depending on aspects of the microstructure, such as pore density and pore size. In our samples, SPPs can be excited by two different wavelengths: (i) via the direct laser field through in-coupling at pore sites [²] (532nm here), and (ii) via excitons in the 2D semiconductor [³]. In WSe₂, both neutral (X₀) and charged excitons (X_T) have a temperature-dependent energy that varies from approximately 750nm to 710nm from room temperature down to 4K [⁴]. The Au-induced doping in WSe₂ will result in higher emission intensity from X_T (versus X₀) and therefore, we can reasonably assume the most energetic and intense exciton dipole energy is at approximately 727nm at 4.5K [³]. (Note: We have updated our assessment of the WSe₂ exciton energy, which we originally assigned at 750nm, but now are assigning at 727nm, which more realistically reflects the primary exciton population at 4.5K for doped WSe₂).

1.1 As authors wrote in the introduction (at the end of the 1st paragraph) ‘In each of these cases, the quality of the 2DC/metal interface will play a central role in determining the ultimate performance of the device’. Therefore, the energy X₀ and X_t emissions are sample specific influenced by the 2DC-metal interface. As such interface is crucial and enabling aspect of presented results, I highly suggest that instead of quoting the value of 727nm, which might be unrepresentative for OPEN-2DC, authors would directly measure the PL energies of X₀ and X_t and use it for the purpose of fitting.

If SPPs are responsible for propagating energy across the film, the total PL decay should be described by

$$I = I_b + I_{532nm} \frac{1}{x} e^{-\frac{x}{\gamma_{532nm}}} + I_{727nm} \frac{1}{x} e^{-\frac{x}{\gamma_{727nm}}} \quad (1)$$

where λ refers to the exciting wavelength, I_λ is the initial wavelength specific intensity at the launching site, γ_λ is the wavelength specific decay length, and I_b is the background signal. The additional $1/x$ term arises from SPPs radiating in all directions in the 2D plane [³]. To gain insight into the most efficient transmission pathways in the OPEN film, we apply the above equation to the most intense PL data from the single-photon emitter (SPE) from Figure 4f in the main text (Figure R1 below). The most intense PL points represent the highest efficiency pathways for energy in-coupling, SPP propagation, and energy out-coupling. The excellent agreement between Equation 1 and the most intense PL data— over three orders of magnitude— provides high confidence that the PL intensity can be modeled by conventional SPP decay for the remotely excited SPE site in Figure 4 of the main text.

1.2 Please define the most intense PL data in the main text, i.e. integrated intensity, peak intensity, over what wavelength?

A good fit of Equation (1) occurs with $\gamma_{532\text{nm}} = 1\mu\text{m}$ and $\gamma_{727\text{nm}} = 15\mu\text{m}$. This implies that the pores do not catastrophically quench SPP propagation. Specifically, the fitted decay length $\gamma_{532\text{nm}} = 1\mu\text{m}$ is consistent with decay dominated by Ohmic losses (i.e., pore scattering is not a significant contributor to SPP decay; calculations in revised SI and Fig. R6 below). On the other hand, if one maintains the assumption of only Ohmic losses and examines decay at $\lambda = 727\text{ nm}$, analytical treatments predict $\gamma_{727\text{nm}} = 32\mu\text{m}$. Our fitted value ($\gamma_{727\text{nm}} = 15\mu\text{m}$) is 53% smaller, implying that there are additional SPP scattering mechanisms, likely from the pores. This implies the OPEN film structure does contribute additional SPP scattering/loss but that this occurs most significantly for longer wavelengths. The fitted decay length here for $\gamma_{727\text{nm}} = 15\mu\text{m}$, by nature, includes all SPP loss mechanisms, (e.g., Ohmic loss, inhomogenities, pore scattering).

Figure R1: Plot of the most intense PL data taken from the remote PL map in Figure 4b, together with a fit from Equation (1) (red line). The individual components of the fit are labeled on the plot, and the total fit (red line) has a $R^2=0.92$, where $I_b=200$, $I_{532\text{nm}} = 9.6\text{E}5$, $I_{727\text{nm}} = 6\text{E}3$, $\gamma_{532\text{nm}} = 1$, $\gamma_{727\text{nm}} = 15$.

1.3 Please comment in the main text on the experimental origin of I_b .

1.4 A curiosity question, can authors comment in the main text on noticeable local increase (bumps) in the histogram fig4f? Do they also reappear when collection spot is fixed at different locations?

The following section shows additional COMSOL simulations to provide a qualitative assessment on how modeled SPP energy varies across a film with and without pores:

Section 2

We performed 3D COMSOL simulations to obtain a qualitative picture for how modeled SPP decay occurs for a system with similar morphology to the OPEN film structure. Figure R2 shows results from simulations for a 50nm thick Au film on 100nm SiO_2/Si , both with and without pores (model length, width = $15\mu\text{m}$, $1.1\mu\text{m}$; pore diameter = 230nm). The cross-sectional power dependence from these simulations is shown in the bottom plot for both the 532nm and 727nm excitation. In this plot we show a cross-sectional slice adjacent to the pores (dotted white line), where the decay is mostly monotonically decreasing.

The main conclusions from these simulations are: (i) for 727nm excitation, the SPP decay length decreases by approximately 55% (from $\gamma=20\mu\text{m}$ to $\gamma=9\mu\text{m}$) between ‘no pores’ and

'pores' and (ii) for 532nm excitation the decay length is approximately the same between 'no pores' and 'pores' ($\gamma=0.65\mu\text{m}$). This is consistent with our data and with our hypothesis that SPPs at both $\lambda = 532$ and 727nm are present, are seen in the PL response, and, for the case of $\lambda = 727\text{nm}$, are responsible for generating remote PL over $\sim 10^3$ μm .

Figure R2: 3D COMSOL simulation of a 50nm thick Au film on 100nm SiO_2/Si with and without a line of pores (diameter=230nm, spacing=1.1 μm ; model length = 15 μm , model width = 1.1 μm). The power decay of SPP waves at different wavelengths is shown in the top three images. The SPP is launched from the left side of the image and propagates to the right side of the image. The bottom plot shows the normalized power cross-section (taken off center at 350nm from the edge; dotted white line) for 532nm and 727nm excitations, both with and without pores. We added an exponential decay (dashed lines) to show the approximate decay length (γ) for each model.

I appreciate the simulations which shows consistently faster decay rate of SPPs induced by the porous structure for the pump laser which supports the experimental SNOM results.

2.1 For the sake of completeness please add to the fig.r2 a 2D map for $\lambda=532\text{nm}$ without the pores

2.2 Why authors do not include ML WSe2 in the simulations? If the thickness of ML-WSe2 is smaller than the skin depth of propagating SPP one might have to consider a hybrid 3 layer interface (Air/WSe2/Au) along which a characteristic SPP propagates. This would be very insightful information.

2.3 Taking into account point 2.1 please clarify in the main text why it is expected to observe similar scaling down of SPP propagation in both WSe2 and Au? Results for SPP decay in Au are generated by a simulation, whereas the reduced γ_{WSe2} is taken from PL that contains other effects such as in/outcoupling efficiencies. In other words, please explain in the main text why this is a 'good' comparison? Perhaps, one could imagine a control experiment by illuminating OPEN (with and without 2DC) with 727nm laser light to measure directly the extend of SPP propagation along the OPEN (without relying on SPEs emissions as shown in fig.3e)

The following section provides additional information and context for estimating SPP transmission efficiencies across the pores in the OPEN films.

Section 3

It is known that SPPs can transmit their energy across gaps, where the transmission probability depends on the gap distance. In gold films, the transmission rate across a 1 μm gap is $\sim 50\%$, for a 200 nm gap is $\sim 80\%$, and for ≤ 30 nm gaps it approaches 100% [6]. For comparison to our system, we note that the OPEN-Au films do not have perfect ‘gaps’ as studied by Flynn et al. [6], but instead have a continuous conducting surface of gold. From this, it is reasonable to assume the transmission values in [6] serve as a LOWER bound for comparing transmission losses of pores to that of true metal gaps.

In Figure 2 of the manuscript we show an example of the pore distributions with annealing a WS_2/Au sample. In this experiment, we measured the ensemble average pore areas of $0.028\mu\text{m}^2$ (resulting in 8.9% pore area coverage) and $0.154\mu\text{m}^2$ (resulting in 29.6% pore area coverage-- discussed in main text), following annealing treatments of 15 minutes and 360 minutes, respectively. Assuming a simple circular hole shape (area= $1/4*\pi*d^2$), the ensemble average pore diameters (d) in these samples are approximately 190nm and 440nm, respectively. For similar ‘gap’ dimensions from Flynn et al. [6], we could estimate a lower bound for SPP transmission efficiencies at 60-80% across a wide range of OPEN film morphologies.

As labeled in the manuscript and described in more detail here, Figure 4 was acquired using a ‘sandwich’ sample with TWO layers of gold– $\text{Au}_{25\text{nm}}/\text{WSe}_2/\text{Au}_{25\text{nm}}$ – as illustrated in Figure 1a (bottom) in the main text (also see Figure R3(A) and R9.1 below). The average Au thickness is 50nm and the average pore diameter with exposed WSe_2 in this region is $230\text{nm} \pm 88\text{nm}$ (Figure R3(B)), which would correspond to a gap transmission efficiency from Flynn et al. [6] at approximately 80%. In addition, from the pore areal density we can calculate a linear pore density (L_d) of 1.1 pores/ μm . Taken together, to estimate loss associated with linear scattering between points ‘A’ and ‘B’ due to a pore transmission rate, we assume that every ‘ L_d unit’ the intensity drops by a fixed percentage determined by:

$$I(x) = I_0(T^{x*L_d}) \quad (2)$$

Applying our experimental L_d and variable transmission efficiencies (T), we can compare the loss associated with pore scattering and that associated with the known exponential Ohmic losses in the metal itself (i.e., $I = I_0e^{-x/\gamma}$; note: in the linear approximation, we do not apply the 2D-radial spreading term ($1/x$)). As shown in Figure R3(C) below, a pore transmission rate of $T=50\%$ would produce similar loss to that expected from Ohmic loss in gold at $\lambda=532\text{nm}$ excitation.

To estimate the total loss due to an Ohmic component and a pore transmission component we use the product of:

$$I(x) = I_0 \left(e^{-\frac{x}{\gamma}} \right) (T^{x*L_d}) \quad (3)$$

In Figure R3(D), we plot the normalized data from the most intense PL in the ‘ WSe_2 -exciton’ plasmon region since we know that additional loss from the porous metal film occurs at this wavelength. A good overlap of Equation (3) occurs with a pore transmission

rate of approximately 94%. As reasonably expected, the best transmission rate for the average 230nm pores is larger than that measured for a 230nm gap assessed by Flynn et al. [6], where we estimate the ‘pore transmission’ here to be about 18% larger than the ‘gap transmission’ in Flynn et al. [6]. Finally, we reemphasize that this assessment characterizes the most efficient combined process for exciton excitation (in-coupling), SPP excitation/propagation, and exciton re-excitation (out-coupling). From the plot in Figure 4 of the main text, intensity points that fall below these most intense PL data represent less efficient exciton/SPP propagation/exciton pathways in the OPEN film and shorter propagation lengths.

Figure R3: (A) AFM from the region in Figure 4 of the main text showing positions of the central collection spot (labeled ‘collection’), and the position of the excitation spot at 17 μm distance (labeled ‘excitation’). (B) Particle counting via ImageJ software to produce a mask of the pores where WSe₂ is exposed. (C) Plot comparing loss due exponential decay (solid line, $I=I_0e^{-x/\gamma}$) and that for fixed transmission loss (dotted line) from Equation 2. (D) Plot showing normalized data from the WSe₂-exciton plasmon region, together with Equation (3) using different transmission values.

I appreciate this analysis that determines the average pore size that can be meaningfully used in COMSOL simulations.

- 4.2 Please also specify the definition of intensity plotted in fig.r3c,d i.e. integrated/peak intensity, RT/4K, of 2DX0 2DXT or SPEs? I understand that it is the same as for fig4b. Is that correct?
- 4.3 I am trying to find consistency between different gammas that are simulated and measured, between fig.r2 and fig.r3. Why in fig.r3 gamma=32um was used not 9um as in fig2? Is the WSe2 overlayer or the second layer of OPEN responsible for extending the gamma? If yes, then why? I believe that the data from fig.r3d could be also fitted with a single decay gamma. What would this value of gamma be? For the clarity, the fig3d would benefit from a line showing decay with gamma=9um (from fig.r2) and gamma=15um (from fig.4f main text) T=100%.
- 4.4 As the consequence of the question 3.3. please add more intuitive explanation in the main text about along which interfaces the SPP travels as a function of distance?
- 4.5 If I am not mistaken the data from fig. r3d is the same as in fig.4b. If this is the

case, please clarify in the main text why authors use eq.1 not eq.3.

How such short-travelling SPP can excite SPEs that are 17 μm away from the laser spot?

As the Reviewer noted, we used NSOM measurements to directly measure SPP waves launched by 532nm excitation at a pore site. As demonstrated in Ref [13] from main text, it is also possible to launch SPPs via excitons in a capping TMD layer. Rather than emitting from the film vertically as PL, the energy released upon recombination of the exciton is coupled into a SPP mode. Coupling energy from the exciton to the SPP mode is analogous to a dipole on the surface of the metal, which can meet the necessary wavevector conditions to couple light into an SPP mode. Therefore, in our OPEN-TMD films we observe BOTH SPP modes, one directly excited by the incident laser at the pore sites (532 nm here), as well as those excited through exciton recombination in the TMD layer (727nm in WSe₂) at the pore sites.

The total intensity of a propagating SPP wave in a metal film is a combination of its intrinsic decay length (γ) and the power applied to excite the wave (Equation 1). Equally important, γ is also wavelength dependent. The excellent fit of Equation 1 demonstrates that with a decay length of $\gamma=1\mu\text{m}$, intensity from the 532nm SPP is still measurable up to $\sim 6\mu\text{m}$ away. However, the intensity decreases by 3-orders-of-magnitude as expected. As we have shown above, the PL intensity beyond $\sim 6\mu\text{m}$ becomes well described by a decay constant of $\gamma=15\mu\text{m}$. It is this longer decay constant that we attribute to the expected 'exciton-excited' SPP mode, which is at a longer wavelength (727nm) than the excitation laser, and is able to propagate at least $17\mu\text{m}$ to remotely excite the SPE under study.

Thank you for the explanation. I agree with the authors that NSOM measurements confirm the propagation length of SPP launched by 532nm laser pump. For the comment about SPP propagating along WSe2 please see other above comments where decay rates gamma are estimated.

Would cross-correlation ($g_2(\tau)$) measurement between two different 'non local' SPE provide an insight to involvement of SPP? A SPE that is further from the origin of SPP should in principle emit later than a SPE that is closer?

If we understand the question correctly, given the fast propagation time of light (e.g., to propagate 17 μm is: $17\mu\text{m}/c = 5.67 \times 10^{-14}\text{s}$), it is not in our measurement capability to differentiate between different emitters sites excited at different distances by SPPs.

As demonstrated in Zhou Y. et. al. Nature Nanotechnology 12, 856 (2017), SPP couple to dark excitons in WSe₂. *Do authors observe such enhancement of PL emission from dark excitons in monolayer WSe2 enabled by OPEN?*

Section 4

The work by Y. Zhou et al. [5] makes use of a dielectric layer to isolate the WSe₂ from the Ag plasmonic waveguide which doubles as a gate electrode. In this geometry, it is possible to change the doping of WSe₂ between, *n*-type, intrinsic (*i*), and *p*-type. In contrast, the TMD layers in the OPEN-films are in direct contact with the supporting Au substrate and doped as a result. As shown in Figure 2e of the main text, direct connection between the TMD metal induces surface potential fluctuations across the film and variable doping. Therefore, while the prospect for coupling to dark excitons remains, it is currently not possible to independently modulate the TMD doping to explore this mechanism.

To provide further insight into this question, we have also performed additional COMSOL simulations to examine how well the *in-plane* (e.g., neutral excitons) and *out-of-plane* (e.g., dark excitons) dipoles couple into SPP modes of the metal, similar to Zhou Y. et al. [5]. The most important result from these simulations is that the *in-plane* dipoles couple into SPP modes by orders-of-magnitude larger when a pore is present in the metal, as compared to a continuous metal film as studied in [5]. The resulting $|E|^2$ intensity is close to three (3) orders-of-magnitude more intense in the presence of a pore (Figures R4 and R5). This result also highlights that propagating SPP waves will couple much more strongly into dipoles located within pores.

Figure R4: 2D COMSOL simulations showing the field intensity component E_z from an oscillating dipole ('red dot' in schematic), both in-plane and out-of-plane, above a gold surface with and without a pore. The dipole position is labeled relative to the dotted line at the Au surface and is either 10nm above the Au surface ($z=10\text{nm}$) or at the bottom of the pore ($z=-49.5\text{nm}$) just above the SiO₂ surface. The simulation length is $10\mu\text{m}$ with the dipole centered at $x=5\mu\text{m}$. The material thicknesses are all the same as labeled in the top schematic (note: schematic not to scale). The pore diameter is 230nm.

Figure R5: Intensity cross-section taken from the 2D COMSOL simulations in Figure R4 at 50nm above the surface, where the dipole is centered at $x=5\mu\text{m}$. The plot shows the square of the field intensity ($|E|^2$). The presence of the pore results in significant improvement for the in-plane dipole coupling, where $|E|^2$ is close to three orders-of-magnitude larger than a continuous film (left plot) away from the emitter site.

Thank you for the explanation. These simulations are much appreciated.

- 4.1 Please add another scenario for $z=0\mu\text{m}$ which is a representative situation given the AFM scans fig r9.1.**
- 4.2 Please add another scenario where dipole is located next or at the edge of the pore $x=5.115\mu\text{m}$ as the interaction between SPE and SPP is the greatest given the simulation shown in fig r11. It is also one of the locations where WSe2 SPEs are likely to be induced.**

Authors use work mentioned in ref. 13 and 36 to support their interpretation of the data. In ref. 13, it has been demonstrated that SPP can propagate up to $34\mu\text{m}$. This was achieved thanks to a smooth interface of 2D metal-oxide-semiconductor (Ag and WS₂). On the other hand, in this work porous morphology annealed metal films provides many out-coupling sites for SPP due to numerous terminations of metal layer which effectively create sharp dielectric changes between air and metal. From ref. 36: “The attenuation length is affected by the scattering of SPs on the surface roughness, scratches, and inhomogeneities of the metal film or the adjacent medium [therein 15,16]”. Intuitively, this should decrease SPP propagation. Having this in mind I believe that it is unlikely that SPPs propagating along a porous metal are responsible for exciting SPEs at distances on the order of 10s of μm .

As the Reviewer notes, in Ref. [13] of main text, SPPs were estimated to propagate up to $34\mu\text{m}$ in a ‘smooth’ sample consisting of TMD/Al₂O₃/Ag layers. To help readers more quickly compare propagation lengths in different materials, we have added a plot in the SI of calculated propagation lengths for Ag and Au using physical constants from [7]. We also highlight another important difference between our 2DC/metal interface and that in Ref [13] of main text. In Ref [13] they deposit an amorphous oxide on their metal surface, which does not produce a highly-crystalline, well-defined interface. In our OPEN-2DC samples, the ‘reverse epitaxy’ process produces a perfectly crystalline, atomically aligned and atomically smooth interface (Fig R7, R8 and discussion below). The OPEN-2DC film has atomically flat terraces that are separated by metal atomic step bunches. Such an interface should have lower atomic potential fluxuations as compared to an amorphous oxide/metal interface.

Figure R6: Calculated SPP propagation lengths for Ag and Au using optical constants from [7]. The red stars indicate the longest propagation lengths measured in Figure 4 of the main text. Labeled “Ref. 13” and “Ref. 36” are from the main text.

We also agree with the Reviewers interpretation that morphological features of a thin metal film will decrease SPP propagation, where specific features were discussed and analyzed in Ref. [36] of main text. To be more realistic for our system that is why we referenced the assessment of Ref. [36] of main text to help understand reduced SPP propagation due to morphological features in the metal film. Without morphological feature scattering, $\gamma_{727\text{nm}} = 32\mu\text{m}$ in Au versus our fitted value of $\gamma_{727\text{nm}} = 15\mu\text{m}$. As discussed above, our incorporation of the experimental results by Flynn et al. [6] demonstrate that the gaps in our metal films do not add a significant amount of scattering, despite the presence of edges and a physical break in the metal.

The Reviewer makes a final important point here in that they believe it is unlikely that SPPs can travel on the order of 10s of μm in these porous metal films. We stress that it is this ‘unexpected’ nature of our results that makes it significant and that our demonstration of both the ‘reverse epitaxial’ process, together with non-spatially local SPE emission, that makes our manuscript suitably novel for Nature Communications. The additional analysis provided above provides confidence that SPPs can travel on the order of 10s of μm in these porous metal films.

Thank you for providing the context and the estimation for SPP propagation. As I mentioned in the beginning, thanks to these additional estimations and further refinements I think the presented results and their interpretation seem now reasonable.

There is an ambiguity in estimating SPP propagation length. On one hand, authors use the work from ref. 36 to give the length of $15\mu\text{m}$ for 750nm plasmons, where the influence of metal roughness is studied. On the other, authors use a formula for Ohmic losses (ref), (mentioned in the section 8 in the supplementary) which assumes flat and smooth interface to obtain $1\mu\text{m}$ propagation length for 532nm plasmons. In **Fig.4f** Plotted lines for gamma do not reflect well the data. Are three lines fitted values or fixed values? What are the fitted values for decays? *Could authors improve their strategy for estimating, measuring and fitting SPP propagation length?*

Based on the Reviewers recommendation, we have performed a more rigorous analysis of the PL intensity decay as shown and discussed in Figures R1, R2, and R3 above. Decay length values of $\gamma = 1\mu\text{m}$ and $\gamma = 15\mu\text{m}$ produce a strong statistical fit, consistent with the expected decay lengths for plasmons excited by green or NIR light in this material.

We thank the Reviewer for pointing out a potential source of confusion when discussing both the Ohmic loss equation and the assessment discussed in Ref [36]. We state in the manuscript “The propagation length (γ) of an SPP in our system, dominated by resistive losses in the metal, is estimated at approximately $\gamma \approx 1 \mu\text{m}$ when excited by a 532 nm laser field. [33,36]”. We cited BOTH ref. [36] and the ‘Ohmic loss’ equation.

To clarify this point, we note that at shorter wavelength excitations ($< \sim 600\text{nm}$), the SPP propagation length for Au calculated from ref. [36] and that calculated from the Ohmic loss equation begin to converge. This is due to the fact that at lower excitation wavelengths the impact of morphological scattering becomes *less than* that due to Ohmic losses in the metal itself. We also see this trend in our COMSOL simulations (Figure R2), where the decay length for a model ‘with pores’ and ‘without pores’ is approximately the same at 532nm excitation, but different at 727nm excitation.

Thank you for the explanation. See my comments above.

As the skin depth of SPP is larger than the thickness of metal film used, SPP propagation along OPEN would be modulated by the roughness proportional to the given depth of sharply terminated pores (25 nm) and their density. *How the roughness of OPEN film is taken into account to estimate SPP propagation length?*

By applying Equation 1 to the most intense PL data (Figure R1), we extract the longest propagation length γ for the OPEN film. **This γ value includes in it ALL loss mechanisms in the OPEN film, which includes Ohmic loss, roughness, pore transmission, inhomogenities,** etc... We do not need to know *a priori* what the individual loss mechanisms are to measure the propagation length in the film. To gain quantitative insight into what the additional SPP loss might be over that of Ohmic loss due to pore-to-pore transmission, we preformed the assessment in Figure R3.

Please include marked sentence or its meaning in the main text.

For the single-photon emitter studied in Figure 4, we used a multi-layer ‘sandwich’ OPEN metal structure (Au/WSe₂/Au), where the starting Au thickness of each layer was approximately 25nm. The average metal thickness across the sample is approximately 50nm.

To more quantitatively address the Reviewers questions regarding surface roughness, we have included additional high-resolution AFM and STM scans of an OPEN-WS₂ film (Fig. R7 and R8 below). By doing so, we find our 2DC-passivated gold surfaces can be atomically flat for any given Au grain/terrace, with the possibility of single Au steps or step bunching. The measured RMS roughness at the tens-of-nanometer scale (via STM) is $\sim 10\text{pm}$, where we always observe the expected moiré lattice between the TMD and Au (i.e., perfect crystalline interface). From AFM, the rms roughness (R_a) of a single Au terrace (at the micron scale) measure approximately 0.15nm and across multiple Au terraces measures approximately 1nm. This assessment of TMD/Au roughness suggests there should be limited SPP scattering due to the type of surface roughness examined in Ref [36], which did not have atomically flat Au crystallites.

Figure R7: Higher resolution AFM images of a WS₂/Au sample annealed at 300°C. The surface roughness is analyzed for different regions, ranging from individual Au terraces (a) to multi-terrace areas (c). (b) Example of height variation across the surface due to step bunching of monoatomic Au layers.

Figure R8: STM images were taken in a ScientaOmicron LT STM under UHV ($\sim 10^{-11}$ Torr) conditions with LN₂ cryostat cooling to 78K. Constant-current feedback was used for imaging the surface with an electrochemically etched tungsten tip which was gold plated on sputter-cleaned Au(111) on mica. The sample was aligned under the STM tip by an optical window in the thermal shielding and long-range ex situ microscope. Resulting data was analyzed with WSxM for assignment of RMS roughness and lattice highlighting. In both (a) and (c) the moiré pattern formed between MoS₂/Au is clearly observed, indicating an atomically clean interface.

Very insightful data showing the highly crystalline structure of the 2DC-OPEN interface. This could be even more emphasized in the main text.

5.1 Curiosity question: Would the angle between Au and 2DC in this Moire heterostructure affect the properties of propagating SPP?

The SPPs propagate in directions determined by the polarization of the incident beam. If the claims in the manuscript are to be accurate, I would expect that different polarisations should yield to different intensity distribution of PL maps that are fostered by OPEN. *So, why the PL map in fig. 4b does not reflect this property of SPP and instead the PL map seems to have a circular symmetry which one would expect to observe when exciting with a circularly polarised light?*

We agree with the reviewer that SPPs are influenced by the polarization of the incident beam and during our studies we have examined if there is a transferable effect on the resulting PL intensity. As shown and discussed in the text, our NSOM measurements do show the expected polarization dependence for SPPs excited by the incident 532nm beam. However, we have not yet observed differences in the PL intensity or directionality based on polarization information being retained by the SPP-exciton excitation process.

Upon further consideration this is perhaps not a surprising result. **There are two pathways in**

which SPPs are excited in our samples: (i) by the polarized laser field via pores in the metal (532nm excitation) and (ii) by exciton dipoles in the 2D semiconductor. Regarding pathway (ii), polarization information does not appear to be retained in the exciton-SPP excitation process, which was also observed in Ref [13] of main text (see Fig. 2a in Ref. [13]). In addition, our COMSOL simulations show that the coupling intensity of both in-plane and out-of-plane dipoles into the porous metal structure are on the same order of magnitude (Fig. R4, R5).

5.2 The marked sentence is a good example of intuitive explanation of the presented system. I would encourage to add it in the main text.

This makes me think that there might be another mechanism that enables the dispersed excitation of SPEs. An alternative effect could be that the metal surface with its numerous holes, sharp edges and variations in height scatters efficiently excitation light in all directions which could optically and directly excite neighbouring SPEs. This seems plausible to me since the sufficient excitation power for observing SPEs is in the nW range, as shown in fig. 4e. Moreover, to acquire the PL map shown in fig.3e, 4.5mW of pump power was used which is remarkably high amount of power, suggesting that the scattering might play a role. *Could authors attempt to disprove or confirm this hypothesis?*

We agree with the Reviewer that it is important to provide a control experiment to test for, and ultimately exclude, other possible excitation mechanisms.

First, it is important to clarify that both the experimental setup and the TMD materials in Figure 3 and Figure 4 are different. In Figure 3e we collect a fluorescence image (CCD image) of a WS₂ sample that is illuminated with a laser spot at room temperature (4.5mW). It is not a spatial PL map formed by scanning an excitation or collection spot across the surface with a spectrometer/detector. The data in Figure 3e is presented with a truncated LINEAR intensity scale to highlight the weaker fluorescence/PL that occurs away from the excitation spot. The data in Figure 4a was acquired using conventional PL mapping (at 4.5K) of WSe₂ where the laser excitation (10μW) and collection spot are confocal. In contrast, the data in Figure 4b is obtained by fixing the collection aperture at the center spot (black dot) and scanning the excitation across the surface; intensity is plotted on a LOG scale. In Figure 4 the maximum excitation power used was 10μW.

Regarding the possibility of scattered 532nm light directly exciting a SPE emitter:

(Point 1) The sample studied in Figure 4 is a ‘sandwich’ sample of Au/WSe₂/Au, such that the semiconducting layer is not in-line with the top surface. This geometry will significantly reduce the probability that a 532nm scattering event will directly excite the buried semiconductor at a different location. Figure R9.1 shows an example of a high-resolution AFM image of another Au/WSe₂/Au sample demonstrating how the WSe₂ layer is located below the top surface, thereby further protecting it from low-angle scattered light.

Figure R9.1: (A) AFM height image of a 'sandwich' Au/WSe₂/Au sample. (B) Cross-section profile from the red line in (A). (C) Schematic showing the location of the different layers in the structure.

(Point 2) We have performed an additional control experiment at the same region mapped in Figure 4 to examine the extent of scattered 532nm light from the OPEN film structure. In this control, we perform the same measurement as shown in Figure 4b, but use a 532 nm band-pass filter to only collect scattered 532 nm light (and not PL light). In Figure R9.2 below, we observe that the intensity of scattered 532 nm light drops off at a greater rate than the PL intensity starting at approximately 0.5 μm away from the excitation spot. By 3 μm from the excitation spot, the 532nm signal intensity is at the baseline intensity of ~ 200 counts/s.

Figure R9.2: Three different examples of strategies to lithographically define where dewetting occurs, and subsequently where PL emission from pore sites occurs.

Thank you for the explanation and additional data.

Point 1. I agree with the authors that the geometry of double layer OPEN makes it more difficult for scattered light to reach and excite SPEs in WSe₂.

Point 2

5.3 The data in the fig9.2 shows that there is little but still some scattering due to the sample roughness and for some cases the two data sets even overlap partly due to large variability of data for 600nm long pass (red dots). Like myself, do authors see the importance of subtracting the black curve from the data in Fig4f to arrive with more accurate SPP decay rate?

I agree with authors that 200c/s seems like not so much, but it introduces no negligible and nonlinear background to the PL data. If this is the reason for constant $I_b=200$ c/s in the fitting please mention it in the main text.

It will be fully convinced about excluding the excitation via scattered pump laser if authors show the decay in fig r9.2 up to 10um, fit these decays and compare with the PL decay from fig4f.

If it is true that SPPs excite SPEs at neighbouring pores sites, I would expect that the intensity distribution of PL maps from fig.3e and fig.4b should look similar. However, these PL maps appear to look very different. *Why is that? How would a histogram of peak intensity vs distance (as shown in Fig. 4f) would look like for PL map from Fig. 3e? Why not integrated intensity?*

As mentioned earlier, the data from Figure 3e and Figure 4b were collected in different experimental setups using different TMDs (WS_2 versus WSe_2), and are plotted on different scales (truncated LINEAR, versus LOG). In Figure 3e we collect a CCD image (fluorescence image) of a WS_2 sample that is illuminated with a laser spot at room temperature. The data is presented on truncated linear scale so that the weak emission far away is highlighted, while bright emission is saturated on the image. In Figure 4b, the map is formed on a sample consisting of WSe_2 at low temperature (4K) by a scanning excitation beam across the surface with an excitation power of $10\mu W$. The intensity in Figure 4b is plotted on a LOG scale and none of the data points are saturated on the image.

Thank you for the explanation.

In such systems like OPEN- WSe_2 one could expect Purcell enhancement to be relevant and pronounced in SPE. *Do authors observe such effects? Is this surprising that the 'non-local' emitter from fig. 4b,d,e shows even longer decay time than 'local' emitter? Can authors measure Purcell enhancement of some SPEs?*

We thank the reviewer for raising this interesting possibility and agree that the OPEN films are an attractive platform for additional studies on the quantum optics of SPEs in a richly textured, nanoplasmonic environment. The sharp nanoscale metal features and suspended 2D crystal drums present in OPEN materials may be useful in modifying the local field strength / local density of states at the site of a SPE. We are actively pursuing further experiments to look for Purcell enhancement or other relevant light-matter interactions enabled by this new material, and we suspect other researchers may wish to do the same. We have added language to the manuscript to point out the relevance of OPEN films for these potential applications.

In terms of the specific question about the lifetimes we present in this paper, we expect that the observed lifetime (extracted from photon correlation measurements) will depend both on the local density of states present at the emitter location (i.e., Purcell effect) and the detailed photophysics / level structure of the emitter itself. Experimentally, we find that the lifetime of some emitters depends in a complicated way on optical pump power, and we do not expect that the effective pump power will be the same between local excitation at 25 nW and remote excitation from 10 uW at an in-coupling site 17 um away. The difference in

lifetimes apparent from the data in Fig 4c,d may be related to the local/remote excitation mechanism, and it may also scale with the effective power reaching the emitter in each case.

Thank you for the explanation.

2. The great care was taken by authors to explain the fabrication process including re-orientation of crystallographic structure Au porous films and growth dynamics by providing sufficient evidence for the claims stated. However, as authors mentioned, de-wetting of Au film was not stopped by the 2D overlayer unlike the work quoted in ref 24. (Coa, P. et al. *Advanced Materials* 29,1701536, (2017)). This makes the outcome of the annealing process appears to be ambiguous which might lead to difficulties in reproducing the results by other scientists. *Could authors compare more thoroughly their work with Coa et. al. and elucidate about the reasons why in their case de-wetting is not fully suppressed?*

There exist experimental differences between our work at that of Cao et al. One important difference is the use of hydrogen gas during the annealing process, which we explicitly address in the main text. As stated in the main text: “The Au film properties, substrate and annealing conditions were selected to promote metal dewetting at low temperatures even in the absence of the 2DC layer. We find the addition of hydrogen gas to the annealing ambient reduces the onset temperature of dewetting by at least 100°C as compared to Ar alone (Fig. 1b,c), presumably through the reduction of suboxides and/or local carbonaceous pinning sites.” In Cao et al., they perform their experiments using a nitrogen-only ambient. We specifically show in Figure 1b,c the difference in using hydrogen gas during annealing for a Au/SiO₂ sample. In that example, at 300C under Ar-only ambient there is NO dewetting of Au/SiO₂, while at 300C under H₂/Ar ambient there is significant de-wetting.

We show in Figure 2d how the presence of defects in the 2DC layer also impacts the dewetting process. In general, CVD grown layers such as that grown by Cao et al. have more defects than exfoliated layers and the de-wetting results can vary more from sample-to-sample using CVD films depending on the material quality. In Figure 2d we highlight in a dramatic fashion how the underlying Au dewetting process can be moderated from defects in the capping 2DC. There likely exist differences in the crystalline quality between the CVD-grown graphene films in Cao et al., and the exfoliated materials used here (e.g., polycrystalline versus single-crystal, wrinkle and defect density, etc...), which can lead to decreased dewetting as observed in Cao et al.

Thank you for the explanation. Although mentioned in the previous paragraph, when referring to Coa et al. please make it clearer that the difference in the dewetting mechanism is due to the presence of H₂ gas during the fabrication process.

3. The next major concern is applicability of presented results. Authors write in the abstract: “Our results suggest the OPEN film geometry is a versatile platform that could facilitate the use of layered materials in quantum optics systems”.

I am afraid I don't see a clear advantage of OPEN against lithographically defined structures in terms of providing “a versatile platform that could facilitate the use of layered materials in quantum optics systems”. Creating OPEN by annealing can only follow log-normal distribution, as described in the main text (ref. 27-29). In contrast to the presented de-

wetting mechanism, electron beam lithography allows for much more versatile approach to design a porous metal film (size, distribution of pores). Some advances have been made in this direction from coupling the SPE to a plasmonic waveguide (ref 15). Moreover, there is very little ground covered in the manuscript about the potential impact and benefits of OPEN-2DC for quantum optics systems. *Could authors expand on this? Could authors demonstrate that OPEN structure can be created by other means than thermal annealing for example by electron beam lithographically?* I believe this would broaden the appeal of the manuscript.

Our goal in writing the final line of the abstract was to motivate the work in the broader context of developing tools and techniques that will advance capabilities to link quantum emitter sites across a surface. As such, we have modified this sentence to read: “Our results suggest design criteria for metal/2DC systems that could facilitate the use of layered materials in a wide variety of systems involving quantum emitters and plasmonic nanostructures.”

To our knowledge, this is this first demonstration of exciting a quantum emitter site remotely at a distance of $17\mu\text{m}$ in a 2DC/metal system. In this context, what our work does is demonstrate important design criteria that, to our knowledge, have not been discussed in the literature. We believe the key morphological aspects of our process that help enable this long-distance excitation are: (i) the crystallographic alignment of the 2D semiconductor and underlying metal film and resulting perfect crystalline interface, (ii) pores in the metal that allow for coupling between SPPs and exciton dipole emitters in the 2D semiconductor, (iii) avoidance of multi-step lithographic processes and their potential for contamination or material degradation from exposure to resists, wet chemicals, etching processes, etc.

As we have shown in the main text, there exist several strategies to pattern where pores form during the OPEN-2DC film processing. One strategy is to define ‘pinning’ sites for the metal layer so that it cannot de-wet during thermal annealing (Figure 3d main text). Alternatively, it is possible to lithographically modify the 2DC capping layer to define pore formation (Fig. 2d main text). As a third example, we show here (Figure R10 below) the possibility of lithographically etching the substrate to direct where pore forms and hence, where emission sites are located. In this example, we form linear arrays of emitter sites in an OPEN film with a *long-term* goal of incorporating wave-guiding functionality (which is beyond the current scope of work).

Figure R10: Three different examples of strategies to lithographically define where dewetting occurs, and subsequently where PL emission from pore sites occurs.

Thank you for clarifying this point.

4. A demonstration of a novel device based on OPEN-2DC would be much in place, otherwise I have difficulties to appreciate the novelty of this work which is appropriate for Nat Comms.

We agree that the specific demonstration of a novel device is an important long-term goal for the new discoveries reported in our work. To show a pathway forward, we have provided three (3) different examples of how the OPEN film, and subsequent luminescence, can be lithography patterned using different techniques: (i) patterning adhesion layers beneath the metal (Fig. 3d), (ii) introducing defects into the 2DC layer itself (Fig 2d), (iii) performing dewetting on textured substrates (Figure R10).

For this manuscript, we argue that a new device structure is unnecessary for suitability in Nature Communications given the level of detail required to describe these new discoveries. We further stress that in the Reviewers own opinion, they believe that it is unlikely that SPP propagate 10s of microns in these samples. We have provided significant additional information to give confidence in this interpretation. This supports the novelty of the work and is an important reason for publication in Nature Communications. We have demonstrated a new way to form hybrid 2DC/metal systems that support propagating SPPs and allow for in-coupling and out-coupling of energy via excitons in the 2D semiconductors.

I accept authors argumentations.

Could authors include in the main text a broader introduction or discussion about (i) the difficulties in making such long distance plasmonic networks and (ii) advantage of 2DC-OPEN platform compared to a 2D-SPEs coupled to a photonic circuit.

My more detailed comments include:

- What is the coupling efficiency between SPP and SPE?

To help provide a qualitative picture for how SPP energy is entering a pore site, we examine cross-sectional profiles from an individual pore shown in Figure R3. We selected a pore that is 10.5 μm from the launching site to verify that a measurable amount of SPP energy is still available to interact with excitons in the 2D semiconductor. The primary take away from Figure R11 below is that while the SPP power within the continuous metal film is small, significant SPP power does exist within the pore, where there are intensity 'hot spots' both at the top surface (Au/Air) and at the bottom surface (Au/SiO₂) and thereby can interact with a 2D semiconductor that is either suspended above or coating the inside of pore. As discussed in Figure R4 and R5 here, an *in-plane* dipole found within a pore (either at the top or bottom of the pore) can much more efficiently couple into SPP modes in the metal as compared to the geometry of a planar metal surface. We surmise that it is the unique geometry of a pore with embedded 2D semiconductor that facilitates the coupling between SPP and SPEs.

Figure R11: 3D COMSOL simulation showing the 727nm SPP power across an individual pore (diameter = 230nm). This data is taken from the tenth pore shown in Figure R2 and is 10.5 μ m from the launch site.

Very insightful simulation.

Please include scale bars.

Please add in the main text a comment that these simulations suggest that the SPEs located at the pore edges would interact most with the SPP.

- Is there a correlation between location of pores and SPEs? Does the density of pores correlate with the density of SPE in Fig4a?

Yes. As shown schematically in Figure 3a and 3b of the main text, the brightest emission comes from regions where the TMD layer is not directly supported by an underlying gold layer. In the OPEN film geometry this occurs at pore sites. The PL intensity from pore sites is orders-of-magnitude brighter than from the Au-supported regions (e.g., Figure 3c main text). Figure R12 below provides additional qualitative analysis comparing the PL map from Figure 4a with the pore locations identified by AFM. The density of narrow emitters is on the same order as the density of pore sites.

Figure R12: Comparison of PL from a region in Figure 4a (left side) with the identified pores from AFM analysis (middle image). (Right image) Composite image with semi-transparent AFM height image (from Figure R3A) overlaid with the PL map. The absence of PL/pores in the bottom right of the image is due to a hole in the WSe2 layer. Due to the absence of well-defined fiducial marks, the alignment within (+/-) 1-2 microns.

Here the authors show the evidence (fig. r12) supporting the claim that ‘emitter sites

(are) located in the pores'. However, the data is not included in the supplementary materials. I strongly recommend to do so. Additionally, as it is presented now it is very difficult to see the correlation between the topography and the SPEs' emission sites. I suggest that several cross sections from fig r12c would be a good way of showing this relation.

In literature, it has been shown that the SPEs can appear randomly as well as induced at the etches and inside narrow nanostructures. Can authors comment on what is the most likely mechanism of inducing SPEs localization in OPEN platform?

- I would appreciate statistics and data supporting the claim that nearly 100% of times 2D crystal is suspended. What is the % of 2DC coating the inside of a pore. It would be nice to see AFM cross sections in Fig.2a. From the AFM scans it seems like the film's height is getting more textured with annealing time. Does the RMS roughness increases with time as the pores are getting bigger?

Per the Reviewers request, we have provided additional data to support this claim. In this example, we use different microscopies to span from low resolution ($>100\mu\text{m}$) to high resolution ($5\mu\text{m}$). In Figure R13 below we increasingly provide higher magnification images from one region on an OPEN-WS₂ film. Using AFM imaging at $30\times 30\mu\text{m}^2$ area, we can begin to differential between suspended membranes and membranes coating the pores. From this image, we can identify nine (9) non-suspended membranes out of approximately 2,500 pores.

Figure R13: Example showing the formation of thousands of pores in an OPEN-WS₂ films. From the $30\times 30\mu\text{m}^2$ AFM image, we can identify nine (9) WS₂ membranes which coat the inside of the Au pores (red circles).

Per the Reviewers request, we have provided additional data for AFM cross sections in Figure 2a (Figure R14 below).

This additional data is much appreciated. It is clear now.

Figure R14: AFM cross sections taken from Figure 2a of the main text.

Regarding sample roughness, in the images shown in Figure 2a (and Fig. R14 here), the total image roughness factor (R_a) increases with anneal time by ~ 0.6 nm. The measured image roughness (R_a) for 15 minutes: $R_a=3.63$ nm; 60min: $R_a=3.96$ nm; and 360min: $R_a=4.20$ nm. We note that this R_a analysis also includes height variations caused by the slightly depressed WS_2 membranes, in addition to height variations in the Au layer. As a result, these ‘image R_a ’ values are likely higher than that from the Au film itself. As shown in Figure R4 and R5, the individual Au terraces can be atomically flat with step bunching occurring across the surface as metal re-distributes to form pores. Without including pores in the R_a analysis (Fig R4) R_a is closer to 1nm over a few micron area. As seen in the 360min anneal in Figure R14 (Figure 2a main text), the Au grains and terraces increase in size with annealing time, and therefore have fewer step boundaries across the crystalline surface.

This additional data is much appreciated. I am satisfied with the level of detail in quantifying the roughness.

- The impact on this manuscript depends on the quality of the SPE. Could authors include more detailed characterisation of SPE i.e. linewidth, brightness, spectral jitter?

Per the Reviewers request, we have included additional plots in the SI to show more details of the narrow emitters. Figure R15 below shows the region from Figure 4 of the main text, with each individual narrow emission site identified with a red ‘dot’ (Fig. R15(A)). We also show histograms of the emitter peak frequency and brightness, along with representative examples of individual emitters from across the area studied.

Figure R15: Selected narrow emitters from Figure 4a of the main text.

This data is much appreciated.

Please include the experimental scheme i.e. confocal microscopy, CCD micrograph.

Please add the scale bar for the distance in the PL map.

Can authors specify the definition of the intensity in the figs13 and r13 as peak or integrated intensity over what wavelength range please?

Authors show very well spectra with mostly single lines. Is this representative? Please include statistics of the number of emitters (peaks) per site. Very important merit of a SPE is its linewidth. Can authors add the statistics on the linewidths of the emitters observed in OPEN-ML_WSe2 please?

- What are thicknesses of 2D material in each sample?

For most of the annealing studies and spectroscopy results discussed and analyzed here (Figure 2, 3, 4 of main text), we focus on monolayers since, in the case of TMDs, they are direct gap semiconductors and have strong PL. For the image shown in Figure 1d, we present data on a few-layer graphene region. While the exfoliated samples have a wide range in flake thickness, we focus on monolayers to simplify the analysis and discussion.

Thank you for the clarification. Please include ‘monolayer’ for WSe2 in the main text and elsewhere if missing.

- I appreciate the sound of acronym OPEN. However, the full name - orientated pore enabled network, is not intuitive, hard to understand and it describes only a part of the results as it does not include the crucial involvement of 2DC. Other names mentioned in the manuscript are better candidates for example “self-assembled, highly-textured porous metallic framework” mentioned in conclusion or “porous metallic networks”.

We appreciate the feedback on the OPEN acronym. We have modified the acronym slightly from 'Orientated Pore Enabled Network' (OPEN) to 'Oriented Porous mEtallic Network' (OPEN) as suggested. In order to differentiate which 2DC layer is responsible for the 'reverse epitaxial' process, we add the specific 2DC name to the name (i.e., OPEN-2DC). For example, if we use WS₂ to re-crystallize Au to form a porous metallic network, we write 'OPEN-WS₂'.

Appreciated.

I recommend improving the readability of figures: i) labels are not clear to read where there are not boxes behind the text (for example Fig.1d or Fig.1i)

As suggested, we have added a box behind the figure labels in Figure 1.

Appreciated.

Fig.1a the result demonstrates only a single step annealing. In order to include the last graphic (OPEN-Au / 2DC / OPEN-Au / SiO₂) please include the data which exhibits this structure.

As suggested, we have updated the supplemental information to include an AFM image of the Au/ WSe₂/Au / SiO₂ sample studied in Figure 4 of the main text.

Thank you for the clarification. Please refer to fig S20 when describing the fig1a - the fabrication process.

In Fig1a, please add Ar+H2 for the clarity.

Fig.1d Why OPEN has a different contrast than non-capped Au under SEM?

In Fig. 1d, the central OPEN region has a different contrast due to the conducting graphene layer being suspended above the SiO₂ layer within an individual pore. Outside of this region, the localized Au islands are on insulating SiO₂ and the electron beam charges the SiO₂ differently in the absence of the graphene layer.

Thank you for the clarification.

Fig.3c Is this spectrum taken at RT? I suggest adding PL spectrum of WS₂ on SiO₂ for the comparison where there is no quenching due to metal or Au-assisted increase of recombination rate of spontaneous emission at the pores' edges.

All data take in Figure 3 was collected at room temperature. Per the Reviewer's suggestion, we included a new SI Figure comparing WS₂ on the dielectric PMMA (PMMA will have less scattering from substrate charges as compared to SiO₂). There is about a factor of three (3) difference between the OPEN-WS₂ and WS₂/PMMA PL peak intensity.

Figure R16: Comparison of room temperature PL of WS₂ on different substrates.

Thank you for the clarification. Please refer and comment on the WS₂/PMMA emission in the main text.

Fig.3e It is not clear what is plotted. Integrated or peak intensity? What wavelength range? Scale bar, log or linear scale?

Figure 3e is a fluorescence image acquired using a CCD imaging system. It is not a scanned PL map as shown in Figure 4a,b. A notch filter was used to block the 532nm excitation beam and a long pass filter (>600nm) allows fluorescence from the WS₂ layer to reach the CCD. The integrated intensity is shown on a truncated linear scale.

Thank you for the clarification. Please include this information in some form in the main manuscript.

***Include information about the thickness of ML-WS₂**

***Although specified in the caption the Fig3e needs an indicator on the image that the remote emissions were taken at RT.**

***Please specify in the caption how the image was acquired. I am afraid using freestanding word 'image' can mean many things in the context of PL spectroscopy. Please use more specific terminology e.g. Micrograph, CCD integrated image, raster scan. Please include more specific information about this picture in the methods e.g. single shot image, integration time...**

***please add the intensity scale bar**

***In the main text describing fig3e, authors should not describe remote emissions in OPEN-WS₂ sample as 'emitters' unless a spectrum is shown. As authors clarified the micrograph shows the integrated intensity obtained with CCD camera. At RT most of the light originates from 2D excitons XT and X₀ as shown in fig3c**

Fig.4 Are PL maps of a and b cover the same area?

Yes, they measure the same area.

Thank you for the clarification. Please include this in the main text.

Fig. 4a It is not clear what is plotted. Integrated or peak intensity? What wavelength range?

Fig 4a shows the total photoluminescence intensity from each point on the sample as measured by a Si avalanche photodiode placed after a 600 nm long-pass filter. The

wavelength range is defined by the 600 nm filter and the detector's sensitivity cutoff in the NIR. The resulting range is approx. 600 – 1000 nm. We have modified the figure to more clearly identify what was measured.

Thank you for the clarification. Although it is roughly mentioned in the methods section please clarify in the main manuscript how each map was generated. Please also include the setup with CCD in fig S21.

Fig. 4b It is written that the PL map was taken by exciting at the 'dot' location and collecting over entire area including the 'cross' location. What are the excitation are collection locations for g2 data in fig. 4d?

As originally described in the caption, in Figure 4b the collection spot is FIXED at the 'dot' location and the excitation spot is SCANNED to form the image. For Figure 4c, both excitation and collection were at the 'dot' point, while in Figure 4d the COLLECTION was at the 'dot' point and EXCITATION at the 'cross' point.

Thank you for the clarification. To clarify this please modify fig4b the label from 'remote' to 'remote excitation' and fig4a likewise from 'local' to 'local excitation'.

Fig. 4e To obtain the red 'non-local' spectrum, authors write that they used "excitation at the cross and collection at the dot". However, the opposite is true for the PL map in fig. 4b. If the locations are indeed reversed, what is the purpose of this?

The image in Figure 4b is collected as a 'non-local' map, where the COLLECTION spot is held fixed at the 'dot' point, and the EXCITATION beam is scanned over the area.

Thank you for the clarification.

Fig4

***Although mentioned in the caption please include the temperature (4.5K) on the Fig4 a b for the clarity**

***Include information about the thickness of ML-WSe2 and each Au (25nm each)**

***Please add schematics of sandwiched OPEN WSe2 to the figure for the clarity. As in the fig.r3a**

I recognized that owe the authors an explanation. In the first set of comments I wrongly understood that the excitation was fixed at a single position, and the collection was scanned whereas the opposite is true for the presented manuscript. This was due to speckle-like appearance of the intensity map. I assumed that each speckle represents a SPE location. In that case can authors elaborate in the main text on the physical reasons for the intensity profile in fig4b to appear granular instead of showing continuous waveguide-like pathways of efficient SPP propagation, as I would expect?

Can Authors include in supplementary materials the same set of data as in Fig4 just for another emitter on the same sample to show the universality of the SPP assisted excitation scheme?

Only a comment: It would be very interesting to overlay the PL map from fig4b with the AFM scan from fig.r3a to seek a correlation between the most effective pathway and the OPEN's topography.

REFERENCES

- 1 Desai, S. B. *et al.* Gold-Mediated Exfoliation of Ultralarge Optoelectronically-Perfect Monolayers. *Advanced Materials* **28**, 4053-4058, doi:10.1002/adma.201506171 (2016).
- 2 Yin, L. *et al.* Surface plasmons at single nanoholes in Au films. *Applied Physics Letters* **85**, 467-469, doi:10.1063/1.1773362 (2004).
- 3 Shi, J. *et al.* Cascaded exciton energy transfer in a monolayer semiconductor lateral heterostructure assisted by surface plasmon polariton. *Nature Communications* **8**, 35, doi:10.1038/s41467-017-00048-y (2017).
- 4 Wang, G. *et al.* Valley dynamics probed through charged and neutral exciton emission in monolayer WSe_2 . *Physical Review B* **90**, 075413, doi:10.1103/PhysRevB.90.075413 (2014).
- 5 Zhou, Y. *et al.* Probing dark excitons in atomically thin semiconductors via near-field coupling to surface plasmon polaritons. *Nature Nanotechnology* **12**, 856, doi:10.1038/nnano.2017.106 <https://www.nature.com/articles/nnano.2017.106#supplementary-information> (2017).
- 6 Flynn, R. A. *et al.* Transmission efficiency of surface plasmon polaritons across gaps in gold waveguides. *Applied Physics Letters* **96**, 111101, doi:10.1063/1.3360202 (2010).
- 7 Johnson, P. B. & Christy, R. W. Optical Constants of the Noble Metals. *Physical Review B* **6**, 4370-4379, doi:10.1103/PhysRevB.6.4370 (1972).

Taking into account point 2.1 please clarify in the main text why it is expected to observe similar scaling down of SPP propagation in both WSe2 and Au? Results for SPP decay in Au are generated by a simulation, whereas the reduced gamma_WSe2 it is taken from PL that contains other effects such as in/outcoupling efficiencies. In other words, please explain in the main text why this is a ‘good’ comparison?

COMSOL modeling can provide insight into the relative SPP power flow when comparing two different simulation geometries. Given uncertainty in translating details of the experimental samples (e.g., boundary conditions, energy input values, loss channels, etc...) into the simulation model, we do not intend to use these COMSOL results for quantitative assessment of SPP propagation lengths. However, by directly comparing two different simulation models with the same boundary conditions and input parameters, we have more confidence that the relative (qualitative) change between models can be used in a meaningful way to compare to experiment. From this exercise, we find a good qualitative agreement between the experimental difference in SPP decay, and the COMSOL difference in SPP decay using an approximated modeled geometry.

We have updated the main text to emphasize this qualitative agreement between experiment and simulation:

“We have performed COMSOL simulations (SI Fig. S19) to **gain qualitative insight into** how the SPP propagation varies in the presence of pores and find two important results: (i) a decrease in $\gamma_{727\text{nm}}$ of approximately 55% due to pores of similar diameter and density as measured in Fig. 4 and (ii) approximately no difference in $\gamma_{532\text{nm}}$ when pores are included. Both results are in good **qualitative** agreement...”

Please also specify the definition of intensity plotted in fig.r3c,d i.e. integrated/peak intensity, RT/4K, of 2DX0 2DXT or SPEs? I understand that it is the same as for fig4b. Is that correct?

The plot in Fig R3C simply shows two different equations which are temperature independent. It is not experimental data and so does not have a ‘measured’ temperature. The experimental data point shown in Fig R3D is from Figure 4 of the main text.

For the clarity, the fig3d would benefit from a line showing decay with gamma=9um (from fig.r2) and gamma=15um (from fig.4f main text) T=100%.

We have included a gamma=15um line to the decay comparison in the SI.

Please include marked sentence or its meaning in the main text.

We have included the following sentence in the main text:

“Importantly, the γ_λ value includes in it all loss mechanisms in the OPEN film, which include Ohmic loss, roughness, pore transmission, and inhomogeneties.”

Although mentioned in the previous paragraph, when referring to Cao et al. please make it clearer that the difference in the dewetting mechanism is due to the presence of H₂ gas during the fabrication process.

We have added the following sentence in the main text:

“An important difference between our work and that of Cao et al.,²⁴ is the use of hydrogen gas during annealing.”

Please include ‘monolayer’ for WSe₂ in the main text and elsewhere if missing.

We have added “monolayer-WSe₂” to Figure 4 caption, “monolayer-WS₂” to Fig. 2 caption, and “monolayer-WS₂/Au(25nm)/SiO₂” to Figure 3.

In Fig1a, please add Ar+H₂ for the clarity.

We updated Figure 1a by adding “Ar+H₂”

Please refer and comment on the WS₂/PMMA emission in the main text.

We have added the following sentence to the main text:

“When compared to a sample of WS₂ on PMMA, the peak PL intensity of the OPEN-WS₂ film is about three times larger (SI Fig. S10).”

Fig.4 Are PL maps of a and b cover the same area? Please include this in the main text.

We have added “...from the same area...” in Figure 4 caption

4.1 Please add another scenario for $z=0\mu\text{m}$ which is a representative situation given the AFM scans fig r9.1. Please add another scenario where dipole is located next or at the edge of the pore $x=5.115\mu\text{m}$ as the interaction between SPE and SPP is the greatest given the simulation shown in fig r11. It is also one of the locations where WSe2 SPEs are likely to be induced.

We have included the additional simulations in the SI. Notably, the ‘offset’ simulation ($x=5.115\mu\text{m}$), where the dipole is located at the pore edge, shows an additional order-of-magnitude increase in the transmitted SPP energy.

5.2 The marked sentence is a good example of intuitive explanation of the presented system. I would encourage to add it in the main text.

We have replaced a similarly worded sentence in the main text with the suggested sentence from the reviewer response:

Replaced: “Since we expect SPPs excited by two different wavelengths (i.e., the direct laser field and excitons in the 2D semiconductor), the total PL decay should be described by....”

With: “Since there are two pathways in which SPPs are excited in our samples: (i) by the polarized laser field via pores in the metal (532nm excitation) and (ii) by exciton dipoles in the 2D semiconductor, the total PL decay should be described by....”

Although mentioned in the caption please include the temperature (4.5K) on the Fig4 a b for the clarity

We updated Figure 4a,b by adding: “4.5K”, “remote excitation”, and “local excitation”. We updated the caption with: “...(Au/ monolayer-WSe₂/Au/SiO₂).”

Although specified in the caption the Fig3e needs an indicator on the image that the remote emissions were taken at RT.

We included the notation “RT” in Figure 3e.

In the main text describing fig3e, authors should not describe remote emissions in OPEN-WS2 sample as ‘emitters’ unless a spectrum is shown. As authors clarified the micrograph shows the integrated intensity obtained with CCD camera. At RT most of the light originates from 2D excitons XT and X0 as shown in fig3c

We have changed “emitters” to “**emission**” in the discussion of Figure 3.

Please add in the main text a comment that these simulations suggest that the SPEs located at the pore edges would interact most with the SPP. Please include scale bars.

We have added the following sentence to the main text:

“For example, these simulations identify that emitter sites located at the pore edges would interact most strongly with a SPP and it is this dramatically enhanced interaction is likely responsible for our observed SPP-pumped PL.”

We added in the following text to SI Figure S15:

“...(diameter = 230nm, Au thickness = 50nm, SiO2 thickness = 100nm).”

Here the authors show the evidence (fig. r12) supporting the claim that ‘emitter sites (are) located in the pores’. However, the data is not included in the supplementary materials. I strongly recommend to do so.

We have included Figure R12 from the response document into the SI. It is the new SI Figure S21.

Fig 3: Please specify in the caption how the image was acquired. I am afraid using freestanding word ‘image’ can mean many things in the context of PL spectroscopy. Please use more specific terminology e.g. Micrograph, CCD integrated image, raster scan.

We added the descriptor “(integrated CCD image)” to Figure 3 caption